# Predicting sub-population specific viral evolution

**Wenxian Shi**                                                          *wxsh@mit.edu*
*Department of Computer Science*
*Massachusetts Institute of Technology*

**Menghua Wu**                                                          *rmwu@mit.edu*
*Department of Computer Science*
*Massachusetts Institute of Technology*

**Regina Barzilay**                                                    *regina@csail.mit.edu*
*Department of Computer Science*
*Massachusetts Institute of Technology*

**Reviewed on OpenReview:** *https://openreview.net/forum?id=Mae23iEqPS*

## Abstract

Forecasting the change in the distribution of viral variants is crucial for therapeutic design and disease surveillance. This task poses significant modeling challenges due to the sharp differences in virus distributions across sub-populations (e.g., countries) and their dynamic interactions. Existing machine learning approaches that model the variant distribution as a whole are incapable of making sub-population specific predictions and ignore transmissions that shape the viral landscape. In this paper, we propose a sub-population specific protein evolution model, which predicts the time-resolved distributions of viral proteins in different sub-populations. The algorithm explicitly models the transmission rates between sub-populations and learns their interdependence from data. The change in protein distributions across all sub-populations is defined through a linear ordinary differential equation (ODE) parametrized by transmission rates. Solving this ODE yields the likelihood of a given protein occurring in particular sub-populations. Multi-year evaluation on both SARS-CoV-2 and influenza A/H3N2 demonstrates that our model outperforms baselines in accurately predicting distributions of viral proteins across continents and countries. We also find that the transmission rates learned from data are consistent with the transmission pathways discovered by retrospective phylogenetic analysis. The code is available at https://github.com/wxsh1213/vaxseer/tree/main/transmission.

## 1 Introduction

Modeling virus evolution is important for vaccine development and disease monitoring. Specifically, the goal is to predict the change in the distribution of viral variants over time. Current approaches aggregate data across all locations of virus collection, resulting in models that represent monolithic, global distributions of each virus. However, this global landscape is composed of local sub-communities of variants with distinct but interdependent distributions. For example, in January 2021, the most common strain of SARS-CoV-2 was different in Europe (B.1.1.7) and Asia (B.1.1.214), but two months later, the former had spread rapidly across Asia (Fig. 1). Faithfully modeling these dynamics is essential for understanding how the viral landscape evolves locally and designing vaccines effective for each community, especially those with less access to public health resources and are poorly represented by the data.

In this paper, we aim to predict the likelihood of a viral variant occurring at a particular time in specific sub-populations, such as the geographical location where the viral sample was collected. Viral variants are represented by the amino-acid sequence of the primary antigenic protein responsible for infection. A

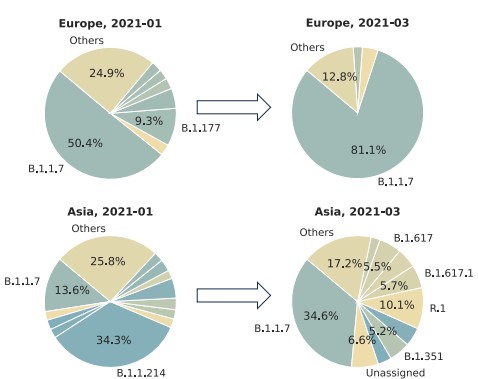

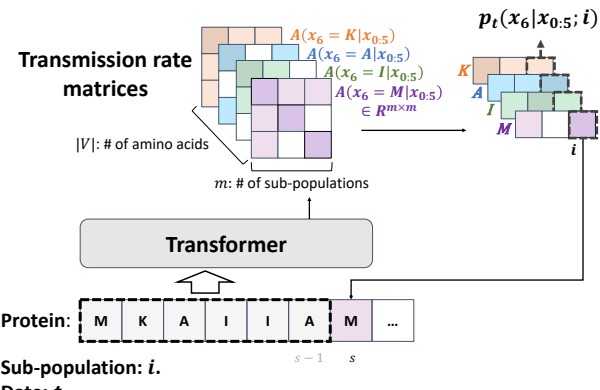

Figure 1: SARS-CoV-2 clade distributions differ by geographical location and change interdependently over time.

Figure 2: The transmission model: an auto-regressive, time-resolved generative model, parametrized in terms of transmission rate matrices.

straightforward approach for modeling location-specific evolution is to separately analyze data for each location, assuming independent evolution. Inherently, this approach leads to data fragmentation, exacerbating the data sparsity problem. Moreover, this approach fails to capture transitions between sub-communities that shape local distributions, especially for highly transmissible diseases. Traditional epidemiology literature has studied transmission dynamics extensively (Sattenspiel & Dietz, 1995; Goel & Sharma, 2020; Balcan et al., 2010; 2009; Colizza et al., 2006), but the hand-crafted rules they employ fall short of capturing the actual complexity of these interactions.

We propose a novel approach to model evolving protein distributions for sub-populations by explicitly capturing the transmissions between them. Specifically, the change in the occurrence of a protein sequence over time across all sub-populations is defined by an ordinary differential equation (ODE), parametrized by a transmission rate matrix related to that sequence. Solving this ODE yields the time-resolved probability of the given protein in different sub-populations, which is determined by the transmission rate matrix and boundary conditions. We leverage language models (Radford et al., 2019) to model the transmission rate matrix and boundary conditions. A practical challenge is that the number of transmission rates scales quadratically with the number of sub-populations, rendering eigen-calculations inefficient for fine-grained sub-populations (e.g. countries). To address this issue, we also propose a hierarchical variation of our model, which leverages the inherent structures that relate sub-populations, e.g. countries can be grouped by their geographical proximity. Specifically, we re-parameterize the transmission rate matrix in terms of intra-group and inter-group transmissions. Both versions of our method are advantageous for sub-populations with limited data, which can benefit from knowledge transferred via the transmission rates with other sub-populations. Moreover, the association defined by the transmission rates can be learned from the occurrences of variants in the dataset, eliminating the need for hand-crafted knowledge.

We evaluate our approach on two viruses, influenza A/H3N2 and SARS-CoV-2, across multiple years, in over thirty countries and all six continents. The evaluation shows that our model outperforms state-of-the-art baselines in predicting future distributions of viral proteins. For SARS-CoV-2, the top 500 protein sequences generated by our model can cover 93.4% of the circulating sequences reported in various countries. Moreover, the transmission rates learned from the data are well aligned with the transmission pathways of viral variants discovered independently by phylogenetic analysis. Finally, our experiments focus primarily on geographical location due to data abundance, but our framework can be easily applied to various types of sub-populations, including individuals of varying ages and vaccination histories, as well as their combinations.

## 2 Method

Let $x = (x^1, ..., x^l) \in V^l$ denote the amino-acid sequence of the viral protein responsible for infections. For example, we model the Hemagglutinin (HA) protein for influenza and the Receptor Binding Domain (RBD)

of Spike protein for SARS-CoV-2. $V$ is the set of amino acids, and $l$ is the length of the protein sequence. Our goal is to model $p_t(x; i)$, the likelihood of protein sequence $x$ isolated at time $t$ in sub-population $i \in S$. $S$ is the set of $m$ sub-populations, i.e., $|S| = m$. Throughout this paper, we use location, defined as the geographical region where the viral sample was collected, interchangeably with sub-population.

## 2.1 Transmission model

The transmission model parameterizes $p_t(x; i)$ through the transmission rate of $x$ between geographical locations. The model outputs $N_t(x)$, the un-normalized probability (occurrence) of $x$ at time $t$ in $m$ locations: $N_t(x) = [n_t(x; 1), ..., n_t(x; m)]$. We assume that the derivative of occurrence over time is a linear transformation of $N_t(x)$, following Kermack & McKendrick (1927); Obermeyer et al. (2022):

$$\frac{dN_t(x)}{dt} = A(x; \theta) N_t(x), \tag{1}$$

in which $A(x; \theta) \in \mathbb{R}_+^{m \times m}$ is called the *transmission rate* matrix of protein $x$. Intuitively, $[A(x; \theta)]_{ij}$ measures the number of people in location $i$ infected by one person from location $j$ during $dt$. In this work, the transmission rate matrix is output by a neural network parameterized by $\theta$ that takes $x$ as input. For concision, we drop $\theta$ in the following equations wherever clear. The ordinary differential equation in Eq. 1 has a closed-form solution,

$$n_t(x; i) = \sum_{j=1}^m u_{ij}(x) \exp(\lambda_j(x)t) \cdot c_j(x); \quad c_j(x) = \sum_{i=1}^m U(x)_{ji}^{-1} \cdot n_0(x; i), \tag{2}$$

in which $\lambda_j(x)$ is the $j$-th eigenvalue of matrix $A(x)$, and $u_{ij}(x)$ is the $i$-th dimension of the $j$-th eigenvector of $A(x)$. $c_j(x)$ is determined by the boundary conditions $N_0(x)$. $U^{-1}$ is the inverse of eigenvector matrix $U$. The derivation can be found in Appendix A.1.

Intuitively, $\lambda_j(x)$ describes the "fitness" of protein $x$ in a "latent" location $j$. Larger $\lambda_j(x)$ indicates that strain $x$ will reproduce faster in the latent location $j$. The eigenvectors combine latent locations with different growth rates $\lambda_j$ and determine the distributions of $x$ in real locations.

To calculate the probability of the sequence $x$ in location $i$ at time $t$, we need to normalize the occurrence $n_t(x; i)$, which is intractable for massive space of sequences. Inspired by autoregressive language models, instead of modeling the occurrence of all sequences, we model the occurrence of the $s$-th amino acid $x_s$ given the prefix amino acids $x_{<s}$:

$$n_t(x_s, x_{<s}; i) = \sum_{j=1}^m u_{ij}(x_s, x_{<s}) \exp(\lambda_j(x_s, x_{<s})t) \cdot c_j(x_s, x_{<s}). \tag{3}$$

The conditional probability of $s$-th amino-acid is calculated accordingly:

$$p_t(x_s | x_{<s}; i) = \frac{n_t(x_s, x_{<s}; i)}{\sum_{x'_s \in V} n_t(x'_s, x_{<s}; i)}. \tag{4}$$

The probability of the whole sequence $x$ can be factorized as the product of the conditional probabilities of each amino acid. As summarized in Fig. 2, given a sequence sampled from location $i$ at time $t$, for each position $s$ in the amino acid sequence, the model takes the previous $s - 1$ amino acids $x_{<s}$ as input and outputs $|V|$ transmission rate matrices of size $m \times m$. The probabilities of the next amino acid in each location (a $m \times |V|$ matrix) are calculated from these transmission rate matrices. For a protein sequence with $L$ residues (amino acids), the model first outputs $L \times |V|$ matrices of size $m \times m$, and then outputs a probability tensor of size $L \times |V| \times m$.

The parameters $\theta$ are trained by the maximum likelihood objective based on tuples $(x, i, t)$ sampled from the training set with empirical distribution $p_{\text{train}}(x, i, t)$. The training loss is

$$\mathcal{L}(\theta) = -\mathbb{E}_{p_{\text{train}}(x,i,t)} \log p_t(x; i, \theta) = -\mathbb{E}_{p_{\text{train}}(x,i,t)} \sum_{s=1}^l \log p_t(x_s; x_{<s}, i, \theta) \tag{5}$$

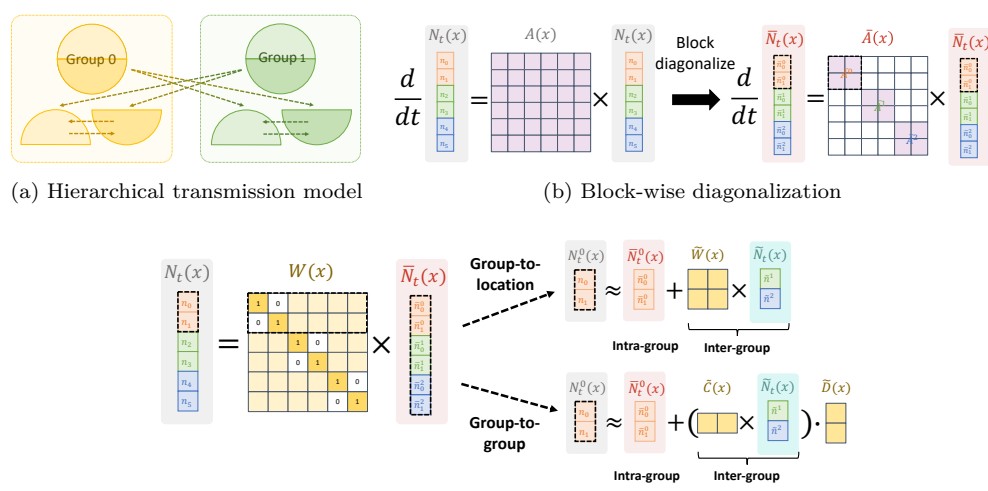

(a) Hierarchical transmission model

(b) Block-wise diagonalization

(c) Inter-group approximation

Figure 3: Hierarchical transmission model. (a) Instead of modeling transmission between all pairs of countries, we model interactions within and across groups of locations (e.g. continents). (b) This idea can be realized by a block-wise diagonal $\tilde{A}$. (c) Two strategies for learning transmissions within each group.

Our model achieves knowledge sharing based on the transmission rate matrix. When the model is trained on a sample from location $i$, the transmission rate matrix is updated through its eigenvectors and eigenvalues (which relate locations with each other). As a result, the probability of that sequence in other locations is also updated. Thus, our model can take advantage of data from different locations to learn the specific distribution for each location.

## 2.2 Hierarchical transmission model

Since eigendecomposition scales as $\mathcal{O}(m^3)$ with the number of locations, applying our model directly to fine-grained locations such as countries becomes highly time-consuming (analysis in Section 4.4). Moreover, the intrinsic structure between sub-populations, such as their geographical proximity, might provide useful inductive biases for modeling their transmissions. Thus, we introduce a hierarchical transmission model that leverages the hierarchical structure of locations to approximate the transmission rate matrix and simplify eigen-calculations. As illustrated in Fig. 3a, we partition locations into groups, e.g. countries to continents. In the hierarchical transmission model, the occurrence in a specific location is calculated by considering transmissions from locations within the same group and from other groups.

We annotate the group of location $i$ as $g_i \in G$, where $G$ is the set of all groups (e.g. continents). We construct a matrix $W(x) \in \mathbb{R}^{m \times m}$, which transforms the transition rate matrix $A(x)$ into a block-wise diagonal matrix. Simplifying $W(x)$ as $W$ and $A(x)$ as $A$, we multiply by $W^{-1}$ on both sides of Eq. 1 to obtain

$$\frac{dW^{-1}N_t(x)}{dt} = W^{-1}AWW^{-1}N_t(x) \quad \longrightarrow \quad \frac{d\bar{N}_t(x)}{dt} = \bar{A}(x)\bar{N}_t(x) \tag{6}$$

in which $\bar{A}(x) = W^{-1}(x)A(x)W(x)$ is a block-wise diagonal matrix with $|G|$ blocks, and $\bar{N}_t(x) = W^{-1}(x)N_t(x)$ is an affine transformation of $N_t(x)$.

Instead of parameterizing the original transmission rate matrix $A(x)$ directly, we parameterize $W(x;\theta)$ and the block-wise transmission matrix $\bar{A}(x;\theta)$. As proved in Appendix A.2, this re-parameterization retains the same representation capacity as the original matrix, provided that $A(x)$ is diagonalizable. Since $\bar{A}(x)$ is block-wise diagonal, Eq. 6 can be further divided into $|G|$ equations: $\frac{d\bar{N}_t^g(x)}{dt} = \bar{A}^g(x)\bar{N}_t^g(x), g \in G$. As illustrated in Fig. 3b, $\bar{A}^g(x) \in \mathbb{R}^{m_g \times m_g}$ is the *transformed* transmission rate matrix corresponding to group $g$ with $m_g$ locations, and $\bar{N}_t^g(x) \in \mathbb{R}^{m_g}$ is the *transformed* occurrence of sequence $x$ in locations within group $g$. Calculating the closed-form solution of $\bar{N}_t^g(x)$ from $\bar{A}^g$ as shown in Eq. 2 is more efficient due to the smaller size of $\bar{A}^g$.

As illustrated in Fig. 3c, $N_t(x)$ can be obtained from the back transformation: $N_t(x) = W(x)\bar{N}_t(x)$. Thus, the occurrence of $x$ in location $i$ is

$$n_t(x;i) = \sum_j W_{ij}\bar{n}_t(x;j) = \sum_{j:g_j=g_i} W_{ij}\bar{n}_t(x;j) + \sum_{j:g_j\neq g_i} W_{ij}\bar{n}_t(x;j), \tag{7}$$

$$= \underbrace{\bar{n}_t^{g_i}(x;i)}_{\text{Intra-group transmission term: } \mathcal{A}} + \underbrace{\sum_{j:g_j\neq g_i} W_{ij}\bar{n}_t(x;j)}_{\text{Inter-group transmission term: } \mathcal{B}} . \tag{8}$$

That is, we rewrite the occurrence of $x$ in $i$ as the sum of contributions from within $i$'s group ($\mathcal{A}$), and from all other groups ($\mathcal{B}$). Here, we assume that $W_{ii} = 1$ and $W_{ij} = 0$ if $g_j = g_i$ and $j \neq i$, which is reasonable because the transformation of elements within groups is redundant during the block diagonalization. We propose two options to approximate $W$: group-to-location (G2L) and group-to-group (G2G). The intuition behind these approximations is that transmission between distant locations is often dominated by some major routines. For example, when traveling from an Asian country to the United States, the journey may involve a stop at a major port in Asia before reaching the destination (group-to-location), or it may involve travel between two major ports across continents before reaching the final destination (group-to-group). We provide the formal description of these approximations below.

**Group-to-location (G2L)**. The first strategy approximates the inter-group transformation weight $W_{ij}$, from location $j$ to location $i$, with $\tilde{W}_{ig_j}$ from group $g_j$ to location $i$ (Fig. 3c). The inter-group transmission term is approximated as

$$\mathcal{B}_{\text{G2L}}(x,i,t) = \sum_{g \in G \setminus \{g_i\}} \tilde{W}_{ig}(x)\tilde{n}_t(x;g), \tag{9}$$

where $\tilde{W}(x) \in \mathbb{R}^{m \times |G|}$ is a linear transformation from groups to locations, parameterized by the neural network. $\tilde{n}_t(x;g) = \sum_{j:g_j=g} \tilde{n}_t(x;j)$ is the sum of the transformed occurrences over locations in group $g$.

**Group-to-group (G2G)**. The second strategy simplifies relations between two locations into a group-to-group term, followed by a term that "distributes" incoming transmissions among locations within a single group. Specifically, we approximate $\tilde{W}_{ig}$ with a transformation $\tilde{C}_{g_i g_j}$ from group $g_j$ to group $g_i$, and a transformation $\tilde{D}_{i,g_i}$ from group $g_i$ to constituent location $i \in g_i$. More specifically,

$$\mathcal{B}_{\text{G2G}}(x,i,t) = \tilde{D}_{i,g_i}(x) \sum_{g \in G \setminus \{g_i\}} \tilde{C}_{g_i g}(x)\tilde{n}_t(x;g), \tag{10}$$

in which $\tilde{C}(x;\phi) \in \mathbb{R}^{|G| \times |G|}$ is the linear transformation from groups to groups, while $\tilde{D}_{\cdot,g}(x) \in \mathbb{R}^{|\{i:g_i=g\}|}$ is the transformation from group to locations belonging to that group.

In both cases, only the group-level $\tilde{n}_t(x;g)$ is required, so for efficiency, we use an auxiliary model to approximate $\tilde{n}_t(x;g)$ instead of computing and summing over individual $\tilde{n}_t(x;j)$. Similar to Eq. 2,

$$\tilde{n}_t(x;g) = \sum_{g' \in G} \tilde{u}_{gg'}(x) \exp(\tilde{\lambda}_{g'}(x)t) \cdot \tilde{c}_{g'}(x). \tag{11}$$

In practice, sub-populations can be clustered into groups based on their geographical coordinates. While these approximations perform well in empirical studies, they may fail to fully capture the complexities of transmission dynamics when certain sub-populations exhibit significantly distinct transmission patterns. This issue can be mitigated by refining the group configurations to better separate such sub-populations.

**Summary.** The occurrence of sequences $x$ in location $i$ from group $g_i$ is modeled as either

$$\begin{aligned} n_t(x;i,\theta) &\leftarrow \bar{n}_t^{g_i}(x;i) + \mathcal{B}_{\text{G2L}}(x,i,t), \text{ or} \\ n_t(x;i,\theta) &\leftarrow \bar{n}_t^{g_i}(x;i) + \mathcal{B}_{\text{G2G}}(x,i,t). \end{aligned} \tag{12}$$

The probability $p_t(x;i,\theta)$ is calculated from $n_t(x;i,\theta)$ using the same autoregressive factorization as Eq. 3. Specifically, given prefix $x_{<s}$ as input:

1. The model outputs the transmission rates $\bar{A}^{g_i}(x_s, x_{<s})$ within group $g_i$, and we compute occurrences $\bar{n}_t^{g_i}(x_s, x_{<s}; i)$ in location $i$ for each $x_s$.

2. The model outputs group occurrences $\tilde{n}_t(x_s, x_{<s}; g)$ and inter-group transformation weights $\tilde{W}_{ig}(x_s, x_{<s})$ (or equivalent $\tilde{C}$ and $\tilde{D}$) for all other groups $g \neq g_i$.

3. The $n_t(x_s, x_{<s})$ is computed combining the intra- and inter-group contributions, as illustrated in Eq. 12. We obtain $p_t(x_s | x_{<s})$ by normalizing $n_t(x_s, x_{<s})$ over all possible values of $x_s$.

The training loss for the hierarchical transmission model is

$$\mathcal{L}(\theta) = \mathbb{E}_{p_{\text{train}}(x,i,t)} \left[ -\log p_t(x; i, \theta) + \lambda \mathcal{L}_{\text{group}}(x, i, t; \theta) \right], \tag{13}$$

in which $\lambda$ is a hyper-parameter and the $\mathcal{L}_{\text{group}}$ is the regression loss for the auxiliary model that predicts the total occurrence within groups:

$$\mathcal{L}_{\text{group}}(x, i, t; \theta) = \left\| \left( \sum_{j:g_j=g_i} \bar{n}_t^{g_i}(x; j, \theta) \right) - \tilde{n}_t(x; g_i, \theta) \right\|^2. \tag{14}$$

## 3 Related work

**Differential equations for modeling epidemiology:** Classic Susceptible-Infected-Recovered (SIR) models (Kermack & McKendrick, 1927; Beckley et al., 2013; Harko et al., 2014) split the population into three sectors, susceptible individuals (S), infectious individuals (I), and recovered individuals (R). Three ODEs describe how the number of infections in each sector changes over time. To model the spatial spread of disease, a series of works (Sattenspiel & Dietz, 1995; Goel & Sharma, 2020; Balcan et al., 2010; 2009; Colizza et al., 2006) introduce mobility operators between different geographical regions into SIR models using transportation data. However, those models focus on the number of infectious for a specific disease. They do not have the capacity to model how different variants evolve over time, which is crucial for vaccine selection and pandemic surveillance. In contrast, our model leverages a neural network to estimate the infection ratio (i.e. probability) of various protein sequences. This flexibility allows our model to predict the probability of any protein variant.

**Predicting fitness for viral variants:** One class of work predicts the fitness of single amino acid mutations and/or predefined clades (Łuksza & Lässig, 2014; Neher et al., 2014; Obermeyer et al., 2022; Maher et al., 2022), but they do not consider fitness based on the whole protein sequence (Gong et al., 2013) and cannot be applied to unseen single amino acid mutations. Another line of work uses protein language models to predict the fitness of protein sequences (Hie et al., 2021; Frazer et al., 2021; Thadani et al., 2023) or whole genomes (Zvyagin et al., 2023). However, they do not explicitly model epidemiological information, which results in a static landscape incapable of capturing changes in the distribution of variants over time. Shi et al. (2023) predicts the distributional changes in variants based on their protein sequences but does not distinguish sub-populations. To the best of our knowledge, our model is the first machine learning model that predicts the temporal distributions of viral proteins in different sub-populations.

**Stochastic Differential Equations (SDEs) and ODEs defined by neural networks**: A series of studies propose generative models based on Stochastic Differential Equations (SDEs) (Song et al., 2021; Lipman et al., 2023; Campbell et al., 2022). These models introduce an abstract time ranging from 0 to $T$ and define a forward (diffusion) process that transforms the data distribution (at time $T$) to a noisy distribution (at time 0). However, these models are unsuitable for our task, as they inherently model a static distribution at time $T$, and the diffusion process does not align with the observed time-reversed evolution process. Other work proposes Neural ODE for representation learning (Chen et al., 2018), which is not directly applicable to our task since the variable evolving in our model is the distribution of discrete sequences, rather than continuous representations.

# 4 Experiments

## 4.1 Settings

**Data processing:** In this paper, we focus on two viruses, influenza A/H3N2 (abbreviated as Flu) and SARS-CoV-2 (abbreviated as Cov). We predict the change in distributions of Hemagglutinin protein (HA) protein for Flu and the receptor-binding domain (RBD) of Spike protein for Cov. We obtain the amino acid sequences of these proteins from GISAID (Shu & McCauley, 2017). Samples collected from human hosts with collection date and collection location information are used in the training and testing. We regard one month (for Cov) or two months (for Flu) as one unit of time.

**Evaluation:** We evaluated models at two levels of geographic granularity — continent and country levels. In multi-year retrospective studies, we employed training and testing splits based on realistic vaccine development processes for both Flu and Cov. For influenza, we emulate the annual recommendation schedule for the northern hemisphere. Since egg-based vaccines require lead times of up to 6 months, we train our models on data collected before February of each year, and evaluate models on the sequences collected from October to March of the next year (winter season), following Shi et al. (2023). To exclude the influence of the COVID-19 pandemic and ensure adequate training data, we evaluated our continent-level model on four influenza seasons, from the 2015 to 2018 winter seasons, and the country-level model on three seasons from 2016 to 2018. In contrast, it takes ∼3 months to produce mRNA vaccines, so we evaluate on Cov in 3-month intervals. Specifically, we trained four models using data collected before four end-points: 2021-07, 2021-10, 2022-01, and 2022-04. For instance, a model trained on sequences collected before 2021-07-01 will be evaluated on sequences collected between 2021-10-01 and 2022-01-01.

We consider six continents (excluding Antarctica) and countries with sufficient data (16 for Flu, 32 for Cov) when training the model. The evaluations are conducted on testing time and locations with sufficient data (10 countries, 5 continents, excluding South America, for Flu; 32 countries, 6 continents for Cov). More details about data pre-processing and statistics can be found in Appendix A.3.

**Metrics:** To measure how well our model and baselines predict future distributions of viral proteins, we used two metrics: negative log-likelihood (NLL) and reverse negative log-likelihood (reverse NLL) (Strudel et al., 2022; Campbell et al., 2024). The NLL is the average negative log-likelihood of observed sequences occurring at the testing time and locations estimated by our model and baselines. The reverse NLL is the average negative log-likelihood of sequences generated by our model and baselines for given testing times and locations, as estimated by oracle models. The oracle models are trained on all available data in each location independently, including sequences in the testing periods. The lower NLL favors models with higher diversity, while the lower reverse NLL favors models with higher specificity. By adjusting the temperature of our models and baselines, we can trade off between diversity and specificity, thereby obtaining the Pareto-frontier in the NLL and reverse NLL space. The Pareto-frontier closer to the lower NLL and reverse NLL corner indicates better model performance. We generated 1500 sequences by our model and baselines with temperatures 0.2, 0.4, 0.6, 0.8, and 1.0. For each model, we obtained the average Pareto frontier by averaging the NLLs and reverse NLLs across testing locations and times with sufficient data. To alleviate the variance caused by the oracle models, we calculated the reverse NLLs by three different oracle models, as illustrated in Appendix A.4.

Inspired by Thadani et al. (2023), we also investigate the ability of our model to predict the prevalent proteins in the future. We generate 100, 300 and 500 sequences with the highest probabilities using beam search (Sutskever et al., 2014) and calculate their *coverage*, defined as the ground-truth frequencies of these sequences occurring in the next three months at different locations. In contrast to the focus of Thadani et al. (2023) on single amino-acid substitutions, we focus on the coverage at the protein sequence level, considering exact matches between the generated sequences and the real-world circulating sequences. We only evaluate performance for Cov due to the sparsity of available data for Flu. We compare the median of coverage across testing times and locations.

**Implementation:** While the transmission rate matrix is not necessarily symmetric, assuming it is a real symmetric matrix is beneficial for training stability and acceleration. Thus, in practice, we parameterize the transmission rate matrix $A_\theta(x)$ as a positive and real symmetric matrix.

For continent-level transmission models, we use a 6-layer GPT-2 (Radford et al., 2019) to parameterize the transmission rate matrix $A_\theta$ and another 6-layer GPT-2 to model the initial occurrence $N_0(x; \theta)$. For hierarchical transmission models, we share the transformer layers for group occurrence, transmission rate within groups, and cross-groups. Adam optimizer with learning rates 1e-5 and 5e-5 are used for FLU and COV, and the models are trained for 80,000 steps for FLU and 30,000 for COV with batch sizes 32 and 256 respectively. The learning rate is linearly warmed up from 0 to the specified value in the first 10% epochs and then decays linearly to zero. More details about training and the hyper-parameters can be found in Appendix A.5.

We use the non-hierarchical transmission model for continent-level tasks and the hierarchical version for country-level tasks. We use the hierarchical loss function but set the number of groups as one at the continent level, equivalent to the non-hierarchical version adding a regularization term. We also present the results for a variant of our transmission model, denoted as `top-3 eig`, where only the three largest eigenvalues are retained for denoising purposes. In hierarchical transmission models, we perform agglomerative clustering based on their geographical coordinates to form groups. We set two groups for FLU and three groups for COV. The countries in each group can be found in Appendix Tab. 6 and Tab. 7. We present the results using both the G2G and G2L approximations.

**Baselines:** We compared our model against three classes of baseline models for both predicting future distributions and generating future prevalent sequences. All baselines were implemented using the same transformer architecture as our model.

1. No location information. **Global model**: a state-of-the-art evolution model (Shi et al., 2023) trained on all data without using the location information. This model can be regarded as a degenerate version of our model with only one location.

2. Baselines that leverage the location-specific data without directly incorporating the location into the model architecture. **Finetune**: fine-tuning the global model on local data. We fine-tune 8 epochs with a learning rate of 1e-5. **LoRA** (Hu et al., 2021): introducing and fine-tuning low-rank adaptations on the global model.

3. Baselines that incorporate location information into the model architecture without considering the interactions between locations. **Prepend Label**: a modification of the global model, prepending a special location token before the amino acid sequences. Both country and continent tokens are prepended for country-level. **Add Embed**: a modification of the global model, adding a location embedding to the hidden state after each transformer layer. **Parameter sharing**: a modification of the global model, sharing the transformer layers but adding different linear output layers for different locations.

We also compared our model with the following non-generative methods, which are applicable only for predicting future prevalent sequences through ranking existing sequences. **Last**: the most frequent sequences occurring within the last 12 months before the end-of-training time in each location. **EVEscape** (Thadani et al., 2023): calculating the fitness for single amino acid substitutions based on a combination of protein language model, structural, and biological information. Note that EVEscape is neither a temporal model nor a sequence generative model. Thus, we utilized it as a scoring metric to re-rank the sequences occurring in the last 12 months in each location. To ensure a fair comparison, we re-trained their protein language models on our training set.

## 4.2 Main Results

As shown in Fig. 4, our model achieves the best frontier in the average NLL and reverse NLL space for both FLU and COV, when predicting protein distributions on both continent and country levels. Compared with the global model, models incorporating location information improve the specificity with lower reverse NLLs, such as Add Embed, Finetune, and Prepend models. However, compared with these baselines which inherently model the evolution in different locations independently, our transmission model achieves even better results in most cases by explicitly considering the transmission between locations. The NLL and

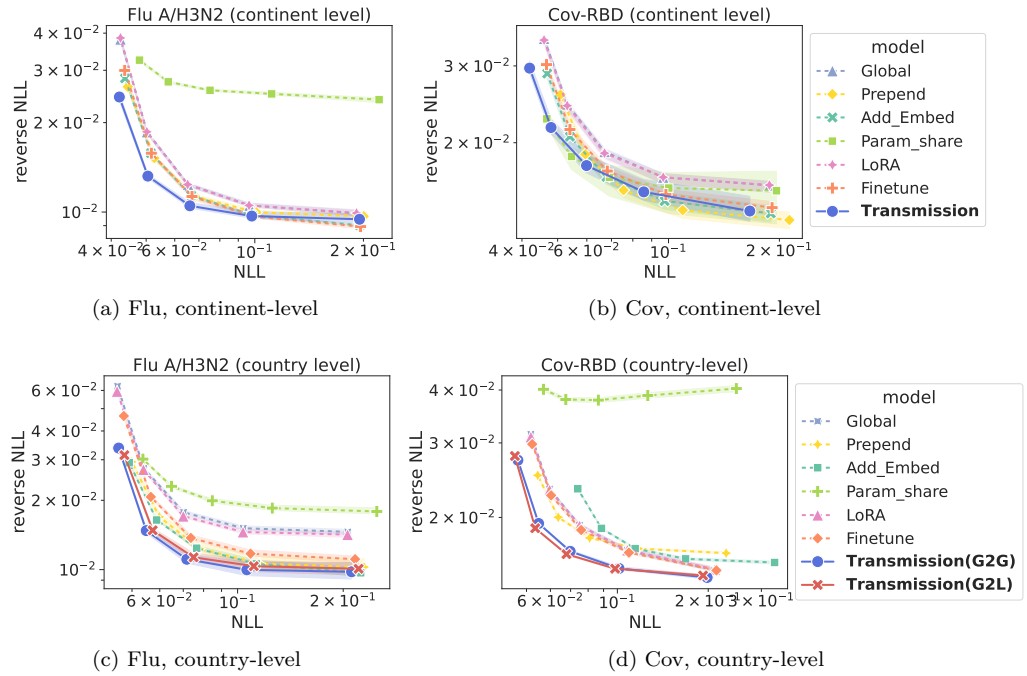

(a) Flu, continent-level

(b) Cov, continent-level

(c) Flu, country-level

(d) Cov, country-level

Figure 4: Average negative log-likelihood (NLL) and reverse negative log-likelihood for FLU and COV. Lower is better. Error bands represent the 95% confidence interval across different oracle models.

reverse NLL for each continent and country can be found in Appendix A.6. As shown in Appendix Fig. 16, our method also achieves the best performance on the worst sub-populations. The exact values of NLL and reverse NLL for Fig. 4 are provided in Appendix Tab. 8-11.

In addition, our model generates more prevalent sequences in the future for COV. As shown in Tab. 1, sequences generated by our model ("Top-3 eig" for continent level and "Hierarchy, G2L" for country level) exhibit the highest coverage. The vanilla "Transmission" model also outperforms other baselines, albeit with slightly lower performance when generating the top 500 sequences. Although "Hierarchy G2G" performs slightly worse than "Hierarchy G2L", it still surpasses other baselines, demonstrating the robustness of our hierarchical approach. The improvement in our country-level model is more pronounced than in the continent-level model, showing that considering the transmission is more critical for fine-grained and interdependent sub-populations. Notably, the top-500 sequences generated by our model can cover 90.9% of circulating viral proteins at the continent level and 93.4% at the country level, three months in advance.

### 4.3 Interpretation of transmission rate matrices as evolution pathways

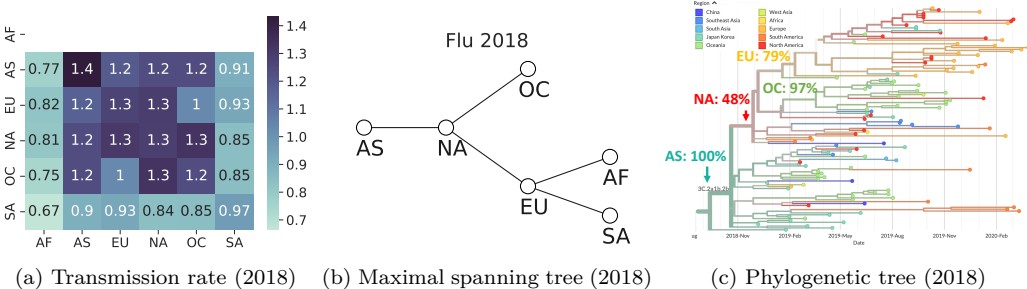

(a) Transmission rate (2018)   (b) Maximal spanning tree (2018)   (c) Phylogenetic tree (2018)

Figure 5: (a) Average transmission rate matrix among sequences collected during the 2018 winter flu season in clade 3C.2a1b.2b. (b) The maximal spanning tree obtained from the rates matrices. (c) The phylogenetic tree obtained from the Nextstrain. Africa (AF) is not included due to insufficient data.

| Model | Continent | | | Country | | |
|---|---|---|---|---|---|---|
| | top-100 | top-300 | top-500 | top-100 | top-300 | top-500 |
| Last (12M) | 0.819 | 0.874 | 0.906 | 0.841 | 0.882 | 0.898 |
| EVEscape | 0.118 | 0.408 | 0.664 | 0.461 | 0.842 | 0.887 |
| Global | 0.828 | 0.855 | 0.862 | 0.853 | 0.882 | 0.884 |
| Prepend | 0.832 | 0.867 | 0.891 | 0.849 | 0.879 | 0.891 |
| Add Embed | 0.828 | 0.875 | 0.889 | 0.853 | 0.879 | 0.894 |
| Param share | 0.816 | 0.870 | 0.889 | 0.818 | 0.870 | 0.884 |
| LoRA | 0.828 | 0.855 | 0.863 | 0.854 | 0.883 | 0.885 |
| Finetune | 0.839 | 0.875 | 0.882 | 0.855 | 0.882 | 0.889 |
| **Transmission** | 0.841 | 0.888 | 0.901 | - | - | - |
| **Top-3 eig** | **0.852** | **0.894** | **0.909** | - | - | - |
| **Hierarchy, G2G** | - | - | - | 0.859 | 0.898 | 0.923 |
| **Hierarchy, G2L** | - | - | - | **0.872** | **0.925** | **0.934** |

Table 1: The coverage (total frequencies of occurrence) of top-100, top-300 and top-500 sequences generated by models for COV. We take the median over testing times and locations.

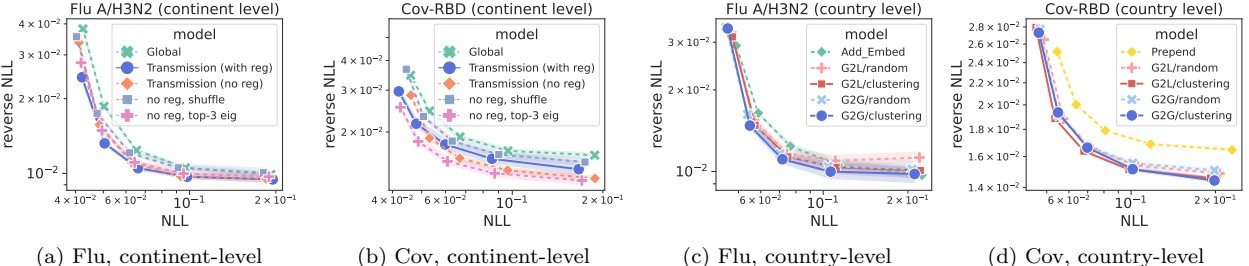

(a) Flu, continent-level    (b) Cov, continent-level    (c) Flu, country-level    (d) Cov, country-level

Figure 6: Ablation study. (a, b) Both the location information and the capacity to factorize the global distribution contribute to the model's performance. (c, d) Similarly, both the inductive biases based on geographic proximity and the ability to model higher-level groups are helpful.

We demonstrate that the transmission rates learned from our method align with the disease spread patterns revealed by phylogenetic analysis (Hadfield et al., 2018), which constructs an evolutionary tree from a set of genetic sequences[1]. Fig. 5a show the average transmission rate matrix of sequences collected in the 2018 winter FLU season and annotated as the clade 3C.2a1b.2b.[2] The transmission rates within continents are larger than those between continents, which implies the reproduction of viruses within continents is faster than across continents. Fig. 5b shows the maximal spanning tree obtained from this transmission rate matrix. This suggests that the clade 3C.2a1b.2b possibly started in Asia, transmitted to North America, and then to Europe and Oceania. This pathway is consistent with the phylogenetic analysis from Nextflu (Fig. 5c). In Appendix A.6, Fig. 14 shows another case of FLU 3C.2a2 clade and Fig. 15 shows the case of COV BA.2 clade. The transmission pathways implied from the transmission rate matrices are also aligned with the phylogenetic trees. It showcases that the transmission rate matrices learned by our method could be used to analyze the transmission pathways and patterns.

## 4.4 Ablation studies

Our ablation studies dissect the value of incorporating sub-populations as additional signals or through architectural changes (factorizing global distributions into mixtures); the runtime and performance trade-off of hierarchical modeling; and other design choices.

First, we demonstrate that the improvement of the transmission model comes from both the incorporation of location information and the advancement of probabilistic modeling. We randomly shuffled the location labels

---

[1]The phylogenic tree is obtained from the Nextstrain: `https://nextstrain.org/seasonal-flu/h3n2/ha`.
[2]Annotated by the nextclade.

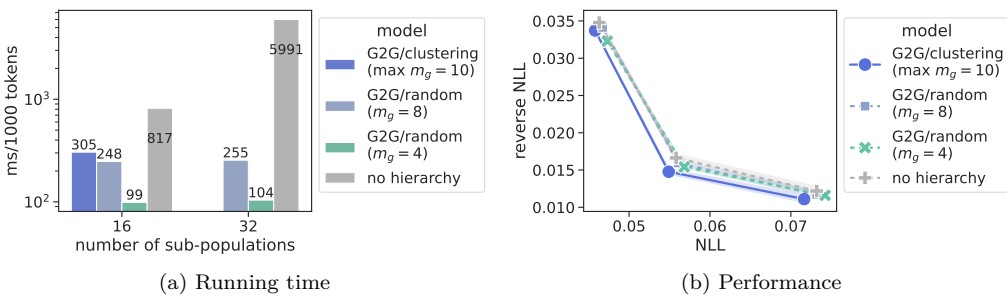

Figure 7: Training time (ms) per thousand tokens and performance for hierarchical transmission model and vanilla transmission model for FLU country-level. $m_g$ is the number of countries in each group. Hierarchical modeling is beneficial for both computing efficiency and performance.

of samples in the training set. In the continent-level results depicted in Fig.6a and Fig.6b, the transmission model trained on randomly shuffled data (annotated as `shuffle`) performs worse. This indicates that our model can effectively utilize location information for more accurate location-specific distribution modeling. However, the model trained on shuffled locations still outperforms the global model, suggesting that the mixture of components with different reproduction rates (represented by different eigenvalues) in Eq. 2 also contributes to the improvement.

We also analyze the impact of grouping locations based on meaningful inductive biases (e.g. geographical proximity), compared to hierarchical modeling in general. To eliminate the influence of group size, we kept the number of countries per group consistent with our hierarchical model but randomly reassigned countries to different groups. As illustrated in Fig. 6c and Fig. 6d, this random assignment (`G2G/random` and `G2L/random`) leads to worse performance, highlighting the advantages of introducing geographical proximity in grouping. However, even with randomly assigned groups, our hierarchical transmission model still outperforms the best baseline model.

Next, we demonstrate that the hierarchical design can improve efficiency significantly while achieving better performance. We compare the training time (milliseconds) per thousand tokens on an NVIDIA RTX A6000 for our transmission model, with and without hierarchy. As shown in Fig. 7, for FLU at the country level, our best hierarchical model, `G2G/clustering`, is 2.6 times faster than the non-hierarchical model while delivering better performance. We also compare the efficiency of hierarchical models with various group configurations. In the `G2G/random` models, we split the countries uniformly and randomly into groups, each containing $m_g$ countries. As expected, the hierarchical model is faster as the number of countries in each group decreases. With $m_g = 4$, the hierarchical model is 8 times faster, achieving performance comparable to the non-hierarchical version. In addition, as the number of total countries increases, the training time for the non-hierarchical model grows cubically. In contrast, the hierarchical model's efficiency is dominated by the number of countries in each group, preventing significant time increases with relatively small group sizes. In practice, our hierarchical model can be further accelerated (up to 8 times faster) by leveraging libraries like cuSOLVER(NVIDIA Corporation, 2023) which optimizes eigen-calculations for smaller matrices.

We studied two additional aspects of our model. We investigated how the number of principal components in the transmission rate matrix affects performance. As shown by the lines annotated as `top-3 eig` in Fig.6a and Fig.6b, if we retain only the largest 3 eigenvalues and corresponding eigenvectors of the transmission rate matrix, the performance remains comparable for FLU but better for COV. This indicates that, in certain cases, performance can be enhanced by de-noising the transmission rate matrix through retaining only the largest eigenvalues. Finally, we found that applying the regularization term $\mathcal{L}_{\text{group}}$ in Eq. 14 slightly improves the performance for FLU, even without employing hierarchical structures (a single group).

## 5 Conclusion

We propose a transmission model to predict the distribution of viral proteins in different sub-populations. By learning the transmission rate between sub-populations based on protein sequences, our model achieves better

performance in forecasting the distribution of Cov and Flu proteins in different locations. Furthermore, the transmission rate matrices learned from the data are aligned the transmission pathways discovered by phylogenetic analysis of given viral strains.

Our method has the following limitations. First, sub-population annotations are required for the training data, which limits the types of applicable sub-populations. Second, we assume that transmission rates remain constant over time, which may not hold if the environment changes unexpectedly. Third, the assumption of a linear relationship between transmission rate and probability in the ODE may oversimplify the dynamics. As part of future work, we plan to incorporate non-linear relationships to address this limitation, such as leveraging a more general formulation of ODEs (Chen et al., 2018) to offer greater capacity.

### Broader Impact Statement

This work provides a tool with significant potential to support local public health efforts. The predictions provided by our model can be used for disease surveillance and vaccine selection in specific countries. However, since the model is trained on historical data, bias during data collection might cause inaccurate results. Decisions based solely on predictive analytics without considering real-world complexities might have unintended consequences. Furthermore, the synthesis and testing of viruses are strictly regulated and controlled by relevant authorities, thereby minimizing potential misuse.

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

# A Appendix

## A.1 Derivation for transmission model

Assuming that transmission rate matrix $A_\theta(x)$ has eigenvectors and eigenvalues, and $A = U\Lambda U^{-1}$. Here, $\Lambda$ is a diagonal matrix, whose $i$-th diagonal element is the $i$-th eigenvalue of $A$. The $i$-th column of $U$ is the $i$-the eigenvector of $A$. Eq. 1 can be written as

$$\frac{dN_t(x)}{dt} = A_\theta(x)N_t(x), \tag{15}$$

$$\frac{dN_t(x)}{dt} = U\Lambda U^{-1}N_t(x) \tag{16}$$

$$\frac{dU^{-1}N_t(x)}{dt} = \Lambda U^{-1}N_t(x) \tag{17}$$

$$\frac{d\bar{N}_t(x)}{dt} = \Lambda \bar{N}_t(x), \tag{18}$$

$$\bar{N}_t(x) = \exp(\Lambda t)\bar{N}_0(x) \tag{19}$$

in which $\bar{N}_t = U^{-1}N_t(x)$. $\exp(\Lambda t)$ is the matrix exponential of $\Lambda t$, which can be easily calculated by taking the exponential in each diagonal element. The $N_t(x)$ can be obtained by

$$N_t(x) = U\bar{N}_t(x) = U\exp(\Lambda t)U^{-1}N_0(x). \tag{20}$$

## A.2 Re-parameterization of hierarchical transmission model

We prove that re-parameterization in the hierarchical transmission model does not reduce capacity. This statement is demonstrated under the setting of two groups, and its extension to multiple groups is straightforward by further partitioning the groups.

**Statement 1.** *If a matrix $A \in \mathbb{R}^{m \times m}$ is diagonalizable, it can be block-diagonalized as $A = W\bar{A}W^{-1}$, in which $\bar{A}$ is a block-diagonal matrix with two blocks.*

*Proof.* If $A$ is diagonalizable, there exists an invertible matrix $U$ and a diagonal matrix $\Lambda$ such that:

$$A = U\Lambda U^{-1}, \tag{21}$$

Let $G \in \mathbb{R}^{m \times m}$ be a block-diagonal matrix constructed as follows:

$$G = \begin{pmatrix} G_1 & 0 \\ 0 & G_2 \end{pmatrix} \tag{22}$$

where $G_1 \in \mathbb{R}^{m_1 \times m_1}$ and $G_2 \in \mathbb{R}^{m_2 \times m_2}$, with $m_1 + m_2 = m$, and both $G_1$ and $G_2$ are invertible. The inverse of $G$ is also block-diagonal, given by:

$$G^{-1} = \begin{pmatrix} G_1^{-1} & 0 \\ 0 & G_2^{-1} \end{pmatrix}.$$

We can write $A$ as:

$$A = UGG^{-1}\Lambda GG^{-1}U^{-1} \tag{23}$$

$$= (UG)G^{-1}\Lambda G(UG)^{-1} \tag{24}$$

We define $W = UG$ and $\bar{A} = G^{-1}\Lambda G$, so that:

$$A = W\bar{A}W^{-1},$$

in which

$$\bar{A} = G^{-1}\Lambda G = \begin{pmatrix} G_1^{-1} & 0 \\ 0 & G_2^{-1} \end{pmatrix} \begin{pmatrix} \Lambda_1 & 0 \\ 0 & \Lambda_2 \end{pmatrix} \begin{pmatrix} G_1 & 0 \\ 0 & G_2 \end{pmatrix} = \begin{pmatrix} G_1^{-1}\Lambda_1 G_1 & 0 \\ 0 & G_2^{-1}\Lambda_2 G_2 \end{pmatrix} \tag{25}$$

is a block-diagonal matrix. □

For any diagonalizable transmission matrix $A$, there exists an invertible matrix $W$ and a block-diagonal matrix $\bar{A}$ such that $A = W\bar{A}W^{-1}$. This re-parameterization preserves the representation capacity of the original transmission matrix.

### A.3 Data pre-processing

### A.3.1 Flu

We downloaded the HA sequences from GISAID which was submitted before 2023-03-02. Sequences without the collection time and location information or from non-human hosts are removed. We also discarded the sequences with less than 553 amino acids. Starting from 2003-10, we discretized every two months as one unit of time. We retained the time with more than 10 samples for each continent.

When training the country-level models, we only considered 16 countries with the most available data. For a specific testing period (for example, from October 2018 to March 2019), we considered the countries and continents with at least 200 viral samples during this period for evaluation, involving 10 countries and 5 continents across all testing periods.

The statistics of data are summarized in Tab. 2 and Tab. 3.

### A.3.2 Cov

We downloaded metadata of COVID-19 from GISAID which was submitted before 2023-11-21. We obtain the RBD sequences by applying the annotated mutations in the metadata to the wild-type Spike protein if the mutations lie between position 319 and 541. Sequences without the collection time and location information or from non-human hosts are removed. Only RBD sequences of length 223 are kept. Starting from 2019-12, we discretized every month as one unit of time. We retained the time with more than 100 samples for each continent.

When training the country-level models, we considered 36 countries with the largest number of samples. For a specific testing period (for example, from January 2022 to March 2022), we considered the countries with at least 1000 viral samples during this period for evaluation, involving 36 countries across all testing periods.

The statistics of data are summarized in Tab. 4 and Tab. 5.

### A.4 Oracle model

We use the evolution model proposed by Shi et al. (2023) as the oracle model, which defines the probability of sequences at time $t$ as

$$p_t(x_s|x_{<s}) \propto \exp(a_\theta(x_s, x_{<s})t + b_\theta(x_s, x_{<s})). \tag{26}$$

The oracle models are trained for each location (continent or country) independently on data across the training and testing time. To mitigate the bias introduced by the oracle models, we trained three oracle models for each location by changing the hyper-parameters.

For FLU at the continent level, we train three oracle models using a 6-layer transformer for $a_\theta$ and another 6-layer transformer for $b_\theta$ with three different random seeds. We trained these models with batch size 32 for 80,000 steps.

For FLU at the country level, we train three oracle models using a 6-layer transformer for $a_\theta$ and another 6-layer transformer for $b_\theta$ with three different random seeds, for 20,000 steps with batch size 32.

For COV at the continent level, we train one oracle model for 100,000 steps with batch size 256 and learning rate 1e-4, using a 6-layer transformer shared by $a_\theta$ and $b_\theta$. We train the second oracle model with the same architecture but for 30,000 steps. The third oracle model is built on one 6-layer transformer model for $a_\theta$ and another one for $b_\theta$, and trained with a learning rate of 1e-4 and batch size 160 for 30,000 steps.

For COV at the country level, we train one oracle model with a shared 6-layer transformer for 30,000 steps with batch size 256 and a learning rate of 1e-4. The second and third oracle models have the same architecture as the first one but are trained by a learning rate of 3e-4 for 20,000 steps with two different random seeds.

### A.5 Hyper-parameters and details of training

We use the base GPT-2 model but with the number of layers as 6 and the hidden size as 768. We introduce a linear layer to output the transmission rate matrices (size $|V| \times m \times m$) from the last hidden state. To make the transmission rate matrix passive, we apply a Softplus function to the outputted transmission rate matrix. We introduce another linear layer to output the boundary condition (occurrence at time $t = 0$, size $|V| \times m$) from the last hidden state, and apply a Softmax function on the vocabulary dimension. We set the $\lambda$ for group regression loss $\mathcal{L}_{\text{group}}$ to 0.1.

We trained our model on 48G NVIDIA RTX A6000. It takes 1 GPU around 18 hours to train a model for FLU at the continent level, and 1 GPU 30 to 36 hours to train a hierarchical transmission model. For COV, training the transmission model on the continent level takes one GPU around 11 hours. Training the hierarchical transmission model takes 2 GPUs around 15 hours.

### A.6 Supplementary results

The NLL and reverse NLL for each continent and country for FLU is shown in Fig. 8 and Fig. 9. The NLL and reverse NLL for each continent and country for COV are shown in Fig. 10 and Fig. 11-Fig. 13.

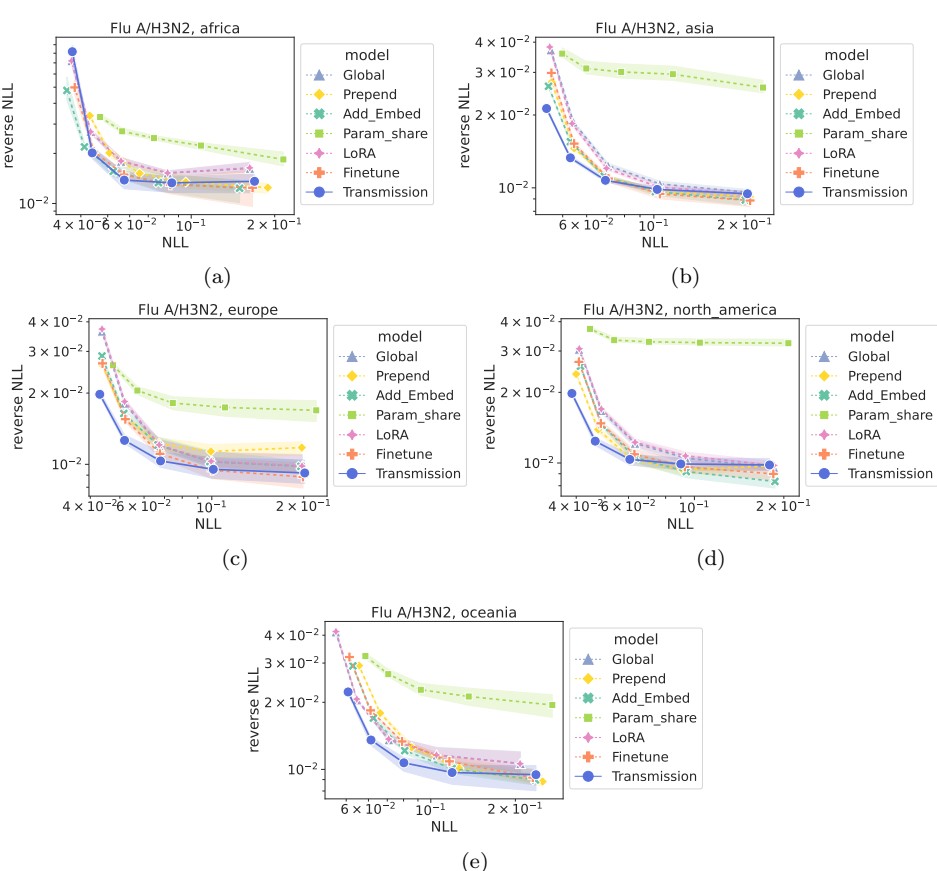

Figure 8: Average NLL and reverse NLL among years for FLU in each testing continent.

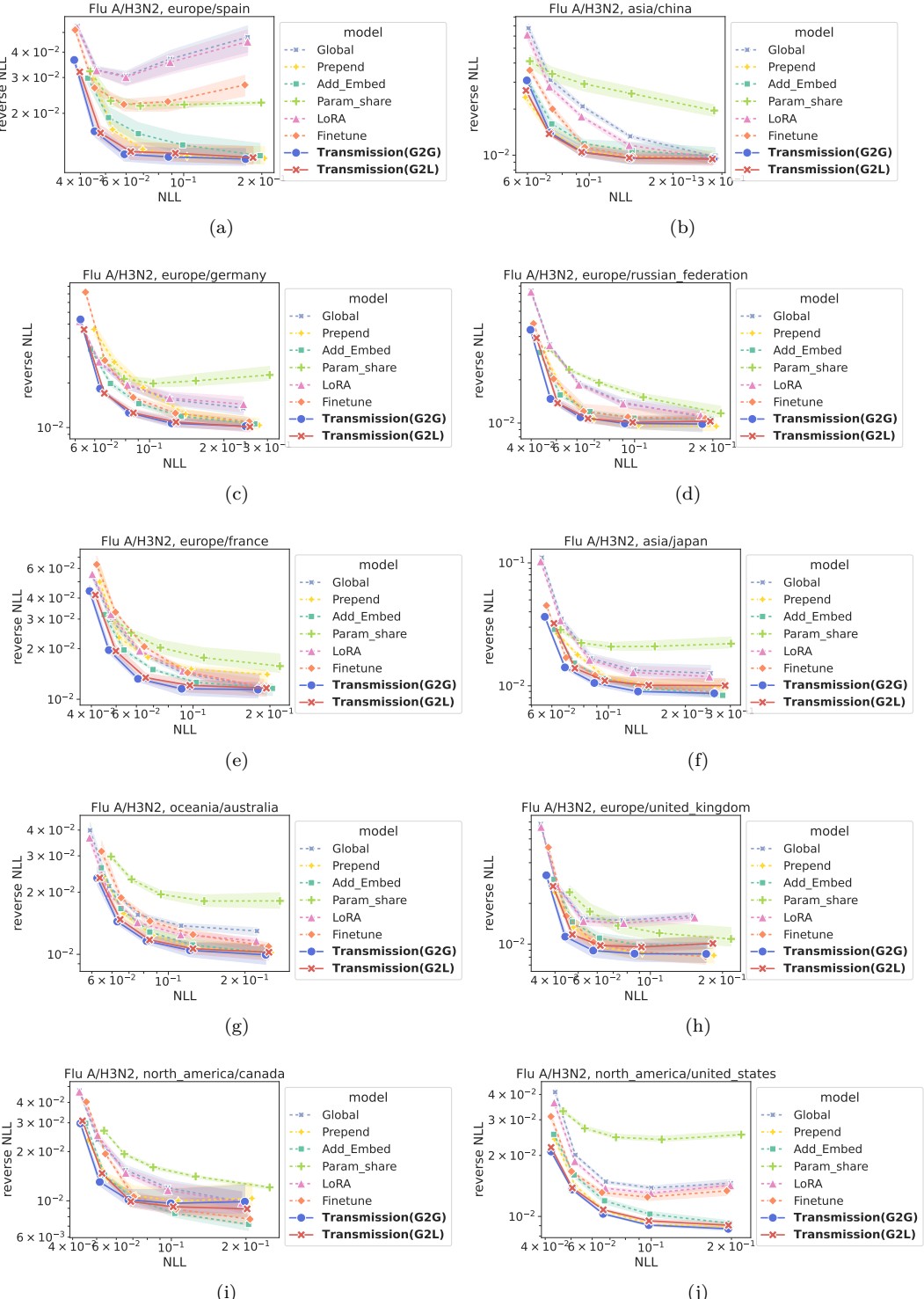

Figure 9: Average NLL and reverse NLL among years for FLU in each testing country.

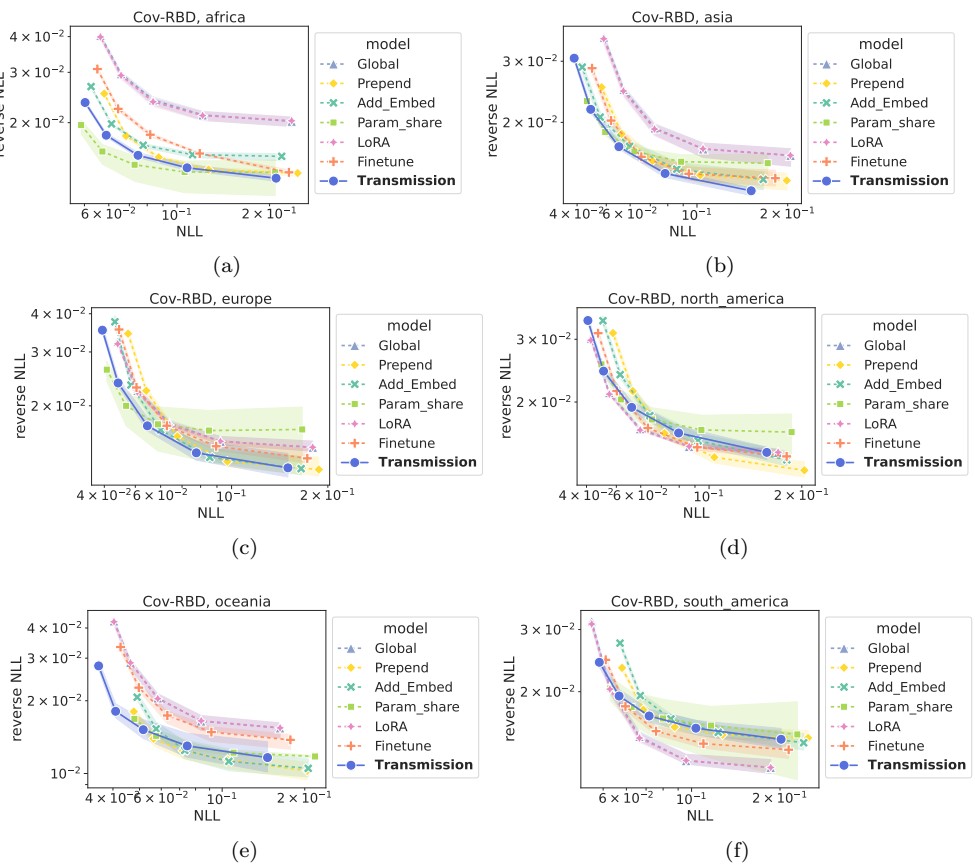

Figure 10: Average NLL and reverse NLL among years for Cov in each testing continent.

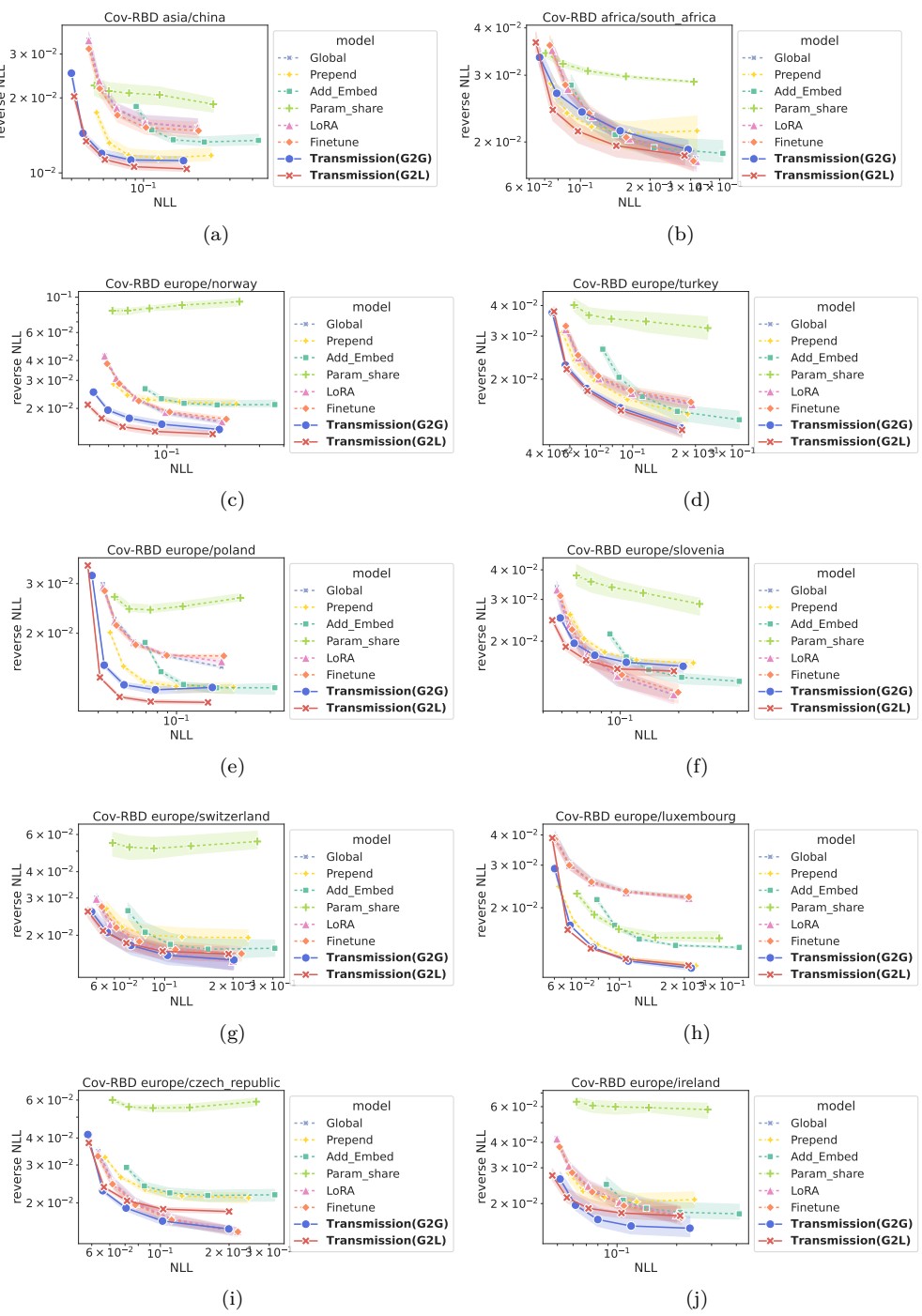

Figure 11: Average NLL and reverse NLL among years for Cov in each testing country (Part I).

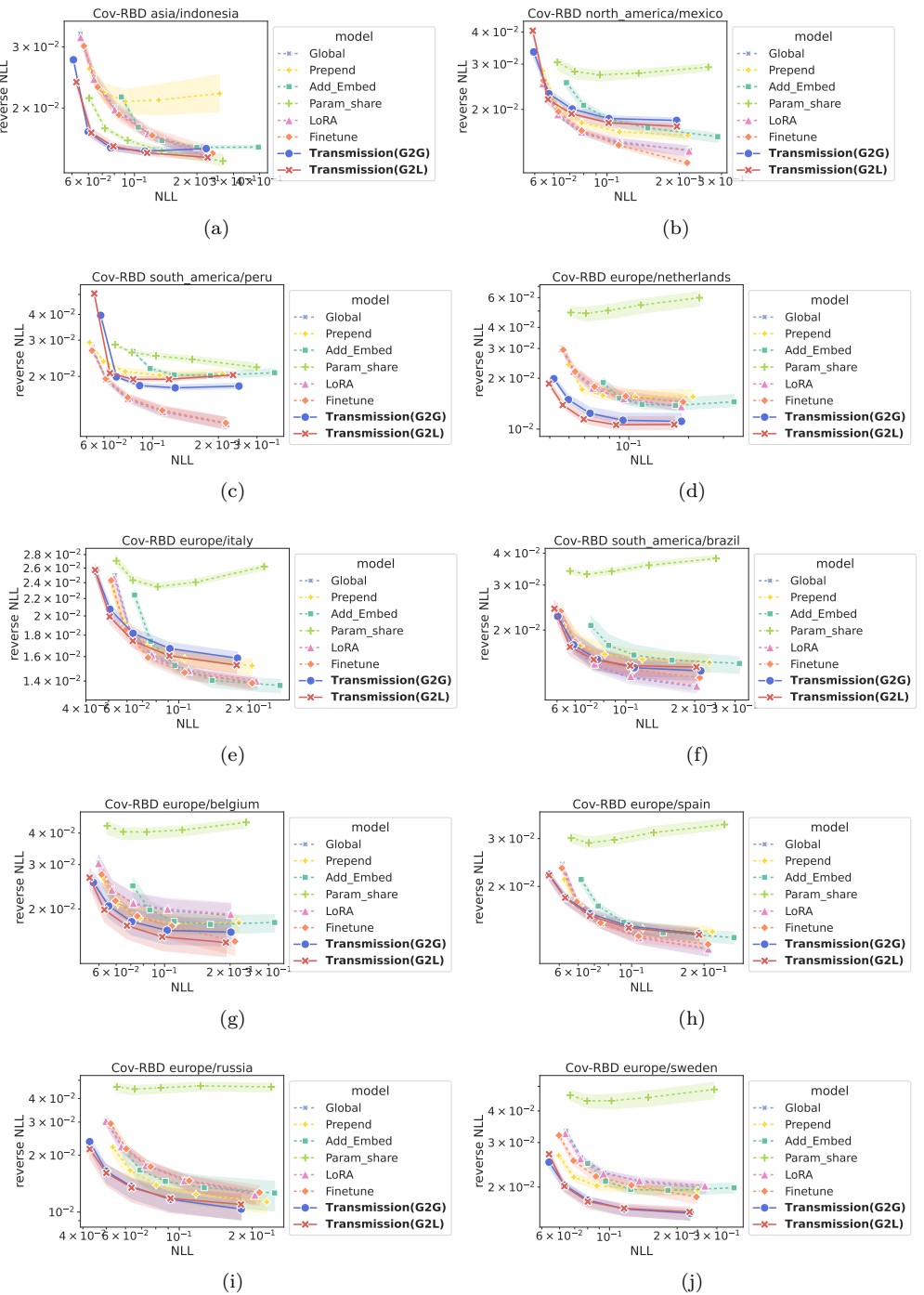

Figure 12: Average NLL and reverse NLL among years for Cov in each testing country (Part 2).

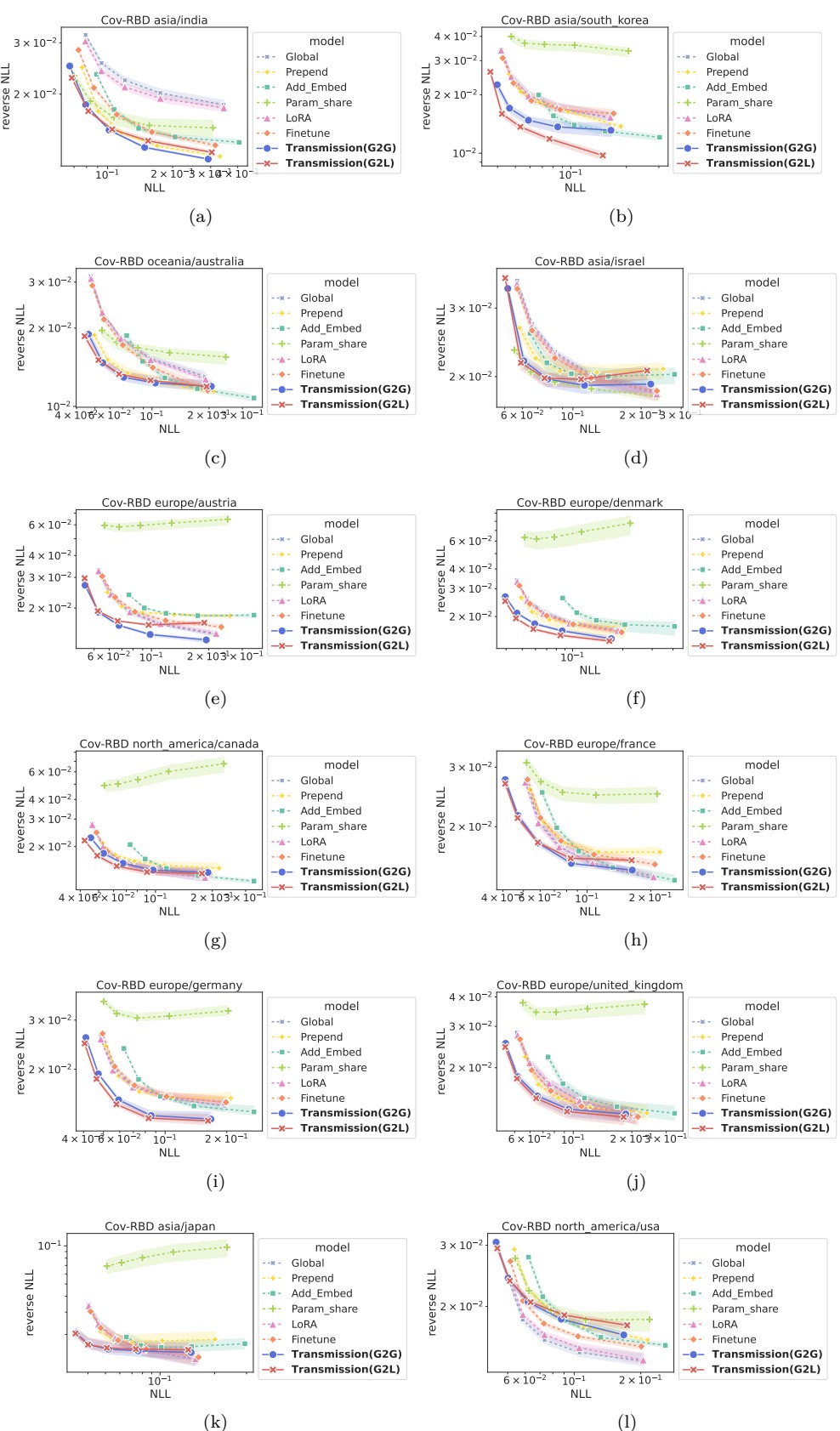

Figure 13: Average NLL and reverse NLL among years for Cov in each testing country (Part 3).

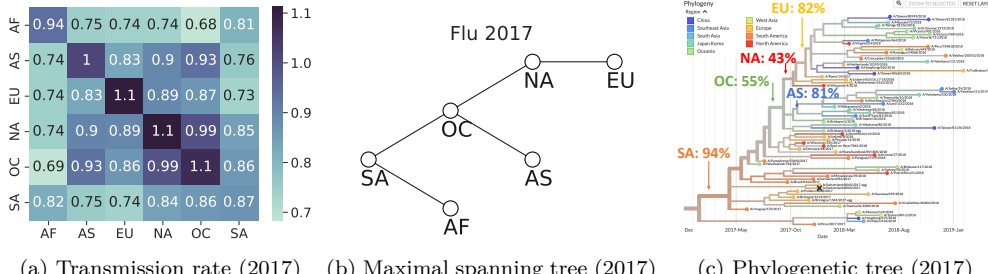

(a) Transmission rate (2017)  (b) Maximal spanning tree (2017)  (c) Phylogenetic tree (2017)

Figure 14: (a) Average transmission rate matrix among sequences collected during 2017 winter FLU season in clade 3C.2a2. (b) The maximal spanning tree obtained from the rates matrices. (c) The phylogenetic tree obtained from the Nextstrain.

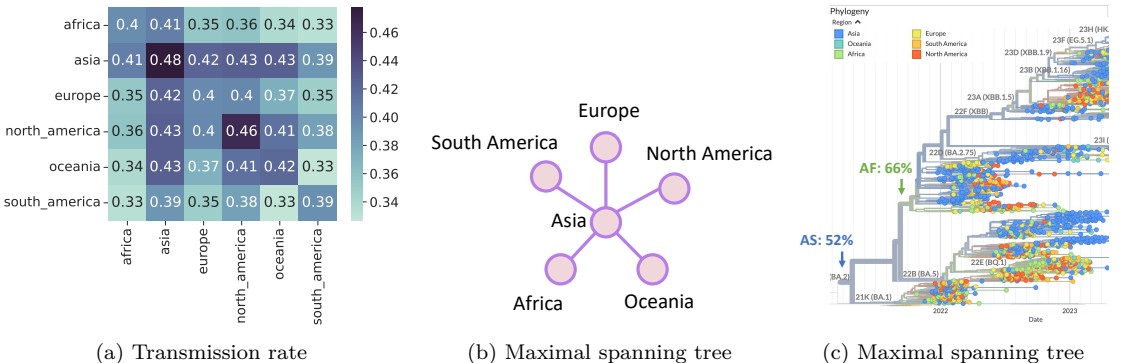

(a) Transmission rate  (b) Maximal spanning tree  (c) Maximal spanning tree

Figure 15: (a) Average transmission rate matrix among sequences collected from 2022-04 to 2022-06 in lineage BA.2 COV. (b) The maximal spanning tree obtained from the rates matrices. (c) The phylogenetic tree obtained from the Nextstrain. Both phylogenetic analysis and our model show that the clade originated from Asia.

| | Number of unique sequences | | | | Number of samples | | | |
|---|---|---|---|---|---|---|---|---|
| | 2015 | 2016 | 2017 | 2018 | 2015 | 2016 | 2017 | 2018 |
| Europe | 1250 | 1492 | 2681 | 3350 | 3116 | 3761 | 7553 | 9576 |
| North America | 2708 | 3142 | 4078 | 5443 | 8315 | 10261 | 13716 | 19740 |
| Asia | 2301 | 2890 | 3687 | 4584 | 5076 | 6361 | 8236 | 10533 |
| South America | 314 | 439 | 503 | 797 | 919 | 1221 | 1339 | 2088 |
| Oceania | 591 | 954 | 1255 | 1718 | 1459 | 2364 | 3482 | 5275 |
| Africa | 354 | 499 | 621 | 792 | 705 | 1017 | 1339 | 1719 |
| Total | 6842 | 8499 | 11550 | 14932 | 19590 | 24985 | 35665 | 48931 |
| Asia / China | 436 | 604 | 721 | 927 | 904 | 1251 | 1478 | 1890 |
| Asia / India | 48 | 48 | 53 | 98 | 92 | 93 | 99 | 168 |
| Asia / Japan | 570 | 683 | 887 | 1099 | 1073 | 1296 | 1705 | 2076 |
| Asia / Singapore | 234 | 270 | 367 | 435 | 561 | 627 | 845 | 1013 |
| Europe / France | 189 | 219 | 385 | 450 | 276 | 327 | 772 | 922 |
| Europe / Germany | 91 | 116 | 254 | 343 | 146 | 178 | 401 | 562 |
| Europe / Italy | 65 | 75 | 145 | 153 | 119 | 144 | 240 | 255 |
| Europe / Netherlands | 68 | 79 | 135 | 187 | 92 | 110 | 200 | 274 |
| Europe / Russian Federation | 184 | 222 | 309 | 379 | 317 | 420 | 647 | 816 |
| Europe / Spain | 109 | 129 | 185 | 230 | 208 | 249 | 372 | 462 |
| Europe / Switzerland | 22 | 25 | 167 | 211 | 36 | 41 | 568 | 733 |
| Europe / United Kingdom | 199 | 225 | 404 | 605 | 533 | 592 | 1088 | 1671 |
| North America / Canada | 327 | 369 | 535 | 763 | 649 | 744 | 1183 | 1801 |
| North America / United States | 2263 | 2617 | 3394 | 4523 | 7182 | 8874 | 11763 | 16808 |
| Oceania / Australia | 467 | 751 | 996 | 1432 | 1097 | 1840 | 2733 | 4366 |
| South America / Brazil | 119 | 192 | 235 | 380 | 232 | 383 | 439 | 752 |

Table 2: Data statistics for Flu in training set.

| | Number of unique sequences | | | | Number of samples | | | |
|---|---|---|---|---|---|---|---|---|
| | 2015 | 2016 | 2017 | 2018 | 2015 | 2016 | 2017 | 2018 |
| Europe | 169 | 1340 | 603 | 1043 | 360 | 4292 | 1782 | 2893 |
| North America | 350 | 1168 | 1122 | 812 | 1123 | 4271 | 4691 | 4026 |
| Asia | 380 | 666 | 459 | 631 | 758 | 1399 | 1028 | 1558 |
| South America | 22 | 100 | 73 | 56 | 36 | 187 | 184 | 162 |
| Oceania | 56 | 197 | 105 | 171 | 89 | 549 | 280 | 527 |
| Africa | 72 | 71 | 41 | 158 | 132 | 162 | 84 | 417 |
| Asia / China | 132 | 106 | 74 | 125 | 274 | 185 | 105 | 215 |
| Asia / Japan | 78 | 200 | 130 | 151 | 138 | 398 | 228 | 369 |
| Europe / France | 17 | 180 | 63 | 155 | 25 | 488 | 108 | 427 |
| Europe / Germany | 25 | 178 | 56 | 84 | 33 | 323 | 84 | 213 |
| Europe / Russian Federation | 8 | 122 | 62 | 124 | 17 | 307 | 156 | 352 |
| Europe / Spain | 5 | 60 | 58 | 84 | 5 | 129 | 125 | 200 |
| Europe / United Kingdom | 16 | 192 | 209 | 74 | 21 | 547 | 673 | 168 |
| North America / Canada | 45 | 217 | 249 | 109 | 91 | 572 | 663 | 324 |
| North America / United States | 291 | 965 | 901 | 699 | 972 | 3573 | 3872 | 3592 |
| Oceania / Australia | 54 | 180 | 98 | 147 | 84 | 479 | 258 | 452 |

Table 3: Data statistics for Flu in testing set.

| | Number of unique sequences | | | | Number of samples | | | |
|---|---|---|---|---|---|---|---|---|
| | 2021-07 | 2021-10 | 2022-01 | 2022-04 | 2021-07 | 2021-10 | 2022-01 | 2022-04 |
| Europe | 2851 | 3980 | 7897 | 16673 | 1.4e+06 | 2.5e+06 | 3.9e+06 | 5.8e+06 |
| North America | 2837 | 3956 | 7214 | 12354 | 997557 | 1.9e+06 | 2.9e+06 | 3.7e+06 |
| Asia | 1503 | 2234 | 3991 | 7496 | 227000 | 416156 | 522226 | 769393 |
| South America | 784 | 1228 | 1962 | 3448 | 88737 | 159608 | 217509 | 285418 |
| Oceania | 205 | 293 | 494 | 880 | 25016 | 45896 | 73489 | 113998 |
| Africa | 763 | 1009 | 1933 | 2704 | 48254 | 75797 | 101557 | 120091 |
| Total | 6041 | 8221 | 15329 | 29042 | 2.8e+06 | 5.1e+06 | 7.7e+06 | 1.1e+07 |
| AF/South Africa | 262 | 391 | 885 | 1275 | 16586 | 26623 | 34622 | 40382 |
| AS/China | 26 | 40 | 50 | 83 | 1638 | 2233 | 2810 | 4248 |
| AS/India | 749 | 1075 | 2333 | 4206 | 67449 | 96526 | 138287 | 190182 |
| AS/Indonesia | 146 | 214 | 322 | 821 | 6502 | 11063 | 13131 | 24446 |
| AS/Israel | 190 | 294 | 631 | 1447 | 13982 | 23934 | 38847 | 70558 |
| AS/Japan | 284 | 571 | 604 | 876 | 83524 | 185183 | 190916 | 278109 |
| AS/South Korea | 145 | 344 | 449 | 655 | 11299 | 22719 | 34588 | 48793 |
| EU/Austria | 220 | 337 | 437 | 884 | 32811 | 51220 | 65025 | 93616 |
| EU/Belgium | 281 | 451 | 713 | 1284 | 33861 | 58923 | 86258 | 127964 |
| EU/Czech Republic | 133 | 185 | 317 | 684 | 8397 | 14115 | 27053 | 41362 |
| EU/Denmark | 340 | 536 | 1659 | 2824 | 119817 | 174989 | 295263 | 470155 |
| EU/France | 492 | 883 | 2048 | 4056 | 57448 | 132715 | 237321 | 361986 |
| EU/Germany | 718 | 1007 | 2057 | 5395 | 150395 | 230313 | 366323 | 599381 |
| EU/Ireland | 132 | 230 | 312 | 487 | 21683 | 38484 | 52945 | 69117 |
| EU/Italy | 433 | 633 | 1019 | 1962 | 45197 | 69711 | 95017 | 120276 |
| EU/Luxembourg | 99 | 136 | 280 | 602 | 11121 | 15183 | 21616 | 31026 |
| EU/Netherlands | 335 | 495 | 666 | 902 | 45127 | 67948 | 91917 | 115284 |
| EU/Norway | 121 | 204 | 288 | 465 | 20707 | 31330 | 45992 | 63759 |
| EU/Poland | 204 | 279 | 478 | 1130 | 18213 | 23484 | 44564 | 82162 |
| EU/Russia | 209 | 304 | 471 | 1139 | 11873 | 17524 | 30290 | 40634 |
| EU/Slovenia | 164 | 227 | 409 | 662 | 21097 | 32613 | 51553 | 67634 |
| EU/Spain | 387 | 619 | 938 | 1822 | 53870 | 87005 | 114204 | 145682 |
| EU/Sweden | 337 | 469 | 784 | 1524 | 89590 | 117489 | 150429 | 193078 |
| EU/Switzerland | 368 | 533 | 882 | 1358 | 46577 | 73502 | 109349 | 137736 |
| EU/Turkey | 215 | 347 | 625 | 1297 | 8659 | 66958 | 80697 | 90021 |
| EU/United Kingdom | 1106 | 1649 | 2874 | 5073 | 557291 | 1e+06 | 1.7e+06 | 2.6e+06 |
| NA/Canada | 405 | 570 | 1000 | 1449 | 112240 | 167621 | 237356 | 306369 |
| NA/Mexico | 294 | 486 | 1021 | 1859 | 20390 | 36595 | 48750 | 61074 |
| NA/Usa | 2578 | 3638 | 6518 | 11207 | 855032 | 1.7e+06 | 2.6e+06 | 3.3e+06 |
| OC/Australia | 171 | 245 | 432 | 782 | 21317 | 39399 | 61434 | 95005 |
| SA/Brazil | 541 | 793 | 1229 | 2462 | 57629 | 98196 | 127431 | 173465 |
| SA/Peru | 101 | 153 | 269 | 487 | 5706 | 10350 | 16421 | 22137 |

Table 4: Data statistics for Cov in training set.

| | Number of unique sequences | | | | Number of samples | | | |
|---|---|---|---|---|---|---|---|---|
| | 2021-07 | 2021-10 | 2022-01 | 2022-04 | 2021-07 | 2021-10 | 2022-01 | 2022-04 |
| Europe | 4873 | 11185 | 6029 | 5440 | 1.4e+06 | 1.9e+06 | 677986 | 489526 |
| North America | 4213 | 6997 | 4920 | 4844 | 976124 | 875411 | 585858 | 513890 |
| Asia | 2262 | 4302 | 3386 | 5308 | 106070 | 247167 | 169727 | 243692 |
| South America | 941 | 1906 | 1088 | 1065 | 57901 | 67909 | 41000 | 39846 |
| Oceania | 240 | 485 | 405 | 636 | 27593 | 40510 | 41305 | 37553 |
| Africa | 1066 | 1106 | 708 | 977 | 25760 | 18534 | 17193 | 10721 |
| AF/South Africa | 540 | 548 | 348 | 240 | 7999 | 5760 | 6566 | 1564 |
| AS/India | 1542 | 2347 | 1795 | 2964 | 41761 | 51895 | 17106 | 22667 |
| AS/Indonesia | 136 | 554 | 311 | 504 | 2068 | 11315 | 5056 | 10019 |
| AS/Israel | 409 | 962 | 832 | 956 | 14913 | 31711 | 32205 | 33706 |
| AS/Japan | 77 | 295 | 257 | 483 | 5733 | 87193 | 63937 | 119228 |
| AS/South Korea | 142 | 228 | 402 | 659 | 11869 | 14205 | 22064 | 24779 |
| EU/Austria | 166 | 514 | 551 | 606 | 13805 | 28591 | 38237 | 42594 |
| EU/Belgium | 357 | 688 | 447 | 394 | 27335 | 41706 | 20057 | 15598 |
| EU/Czech Republic | 171 | 433 | 354 | 399 | 12938 | 14309 | 7277 | 7795 |
| EU/Denmark | 1232 | 1681 | 538 | 344 | 120274 | 174892 | 70330 | 47944 |
| EU/France | 1382 | 2566 | 1576 | 1039 | 104606 | 124665 | 91470 | 49631 |
| EU/Germany | 1311 | 4035 | 2024 | 1643 | 136010 | 233058 | 147251 | 78441 |
| EU/Ireland | 136 | 198 | 167 | 145 | 14461 | 16172 | 15874 | 8820 |
| EU/Italy | 497 | 1176 | 859 | 713 | 25306 | 25259 | 18314 | 14109 |
| EU/Luxembourg | 159 | 395 | 266 | 214 | 6433 | 9410 | 8339 | 7433 |
| EU/Netherlands | 269 | 319 | 263 | 280 | 23969 | 23367 | 14001 | 13988 |
| EU/Norway | 122 | 208 | 166 | 164 | 14662 | 17767 | 6602 | 3477 |
| EU/Poland | 251 | 760 | 108 | 149 | 21080 | 37598 | 1571 | 4001 |
| EU/Russia | 239 | 749 | 149 | 428 | 12766 | 10344 | 1912 | 20857 |
| EU/Slovenia | 220 | 310 | 174 | 173 | 18940 | 16081 | 3822 | 5653 |
| EU/Spain | 433 | 1081 | 807 | 759 | 27199 | 31478 | 27913 | 19124 |
| EU/Sweden | 396 | 920 | 355 | 525 | 32940 | 42649 | 10915 | 20999 |
| EU/Switzerland | 450 | 627 | 246 | 253 | 35847 | 28387 | 6903 | 6659 |
| EU/Turkey | 349 | 745 | 240 | 223 | 13739 | 9324 | 4462 | 3805 |
| EU/United Kingdom | 1723 | 2803 | 1106 | 1027 | 705745 | 893246 | 149120 | 82428 |
| NA/Canada | 537 | 621 | 427 | 511 | 69735 | 69013 | 53255 | 49505 |
| NA/Mexico | 649 | 1093 | 497 | 358 | 12155 | 12324 | 8610 | 11072 |
| NA/Usa | 3777 | 6321 | 4529 | 4501 | 885278 | 785517 | 512906 | 445673 |
| OC/Australia | 222 | 445 | 381 | 579 | 22035 | 33571 | 34465 | 28458 |
| SA/Brazil | 557 | 1478 | 654 | 516 | 29235 | 46034 | 21768 | 14964 |
| SA/Peru | 143 | 268 | 232 | 316 | 6071 | 5716 | 5118 | 11160 |

Table 5: Data statistics for Cov in testing set.

| Group | Countries |
|---|---|
| 1 | Australia, China, India, Japan, Russian Federation, Singapore |
| 2 | Brazil, Canada, France, Germany, Italy, Netherlands, Spain, Switzerland, United Kingdom, United States |

Table 6: Setting of groups for Flu.

| Group | Countries |
|---|---|
| 1 | Austria, Belgium, Czech Republic, Denmark, France, Germany, Ireland, Israel, Italy, Luxembourg, Netherlands, Norway, Poland, Slovenia, South Africa, Spain, Sweden, Switzerland, Turkey, United Kingdom |
| 2 | Australia, China, India, Indonesia, Japan, Russia, South Korea |
| 3 | Brazil, Canada, Mexico, Peru, USA |

Table 7: Setting of groups for COV.

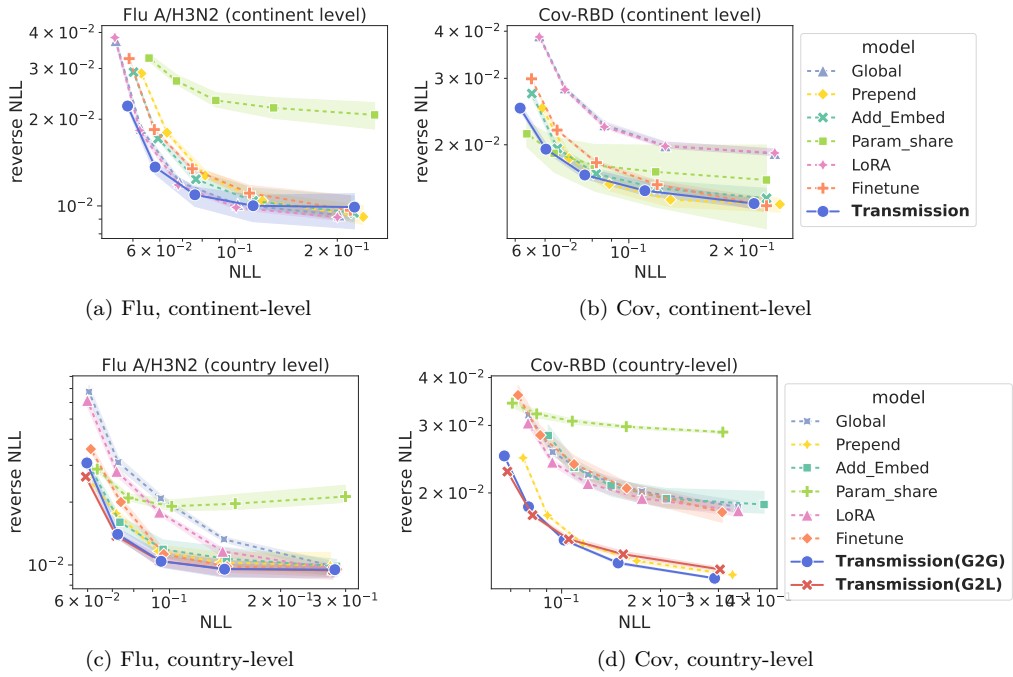

(a) Flu, continent-level

(b) Cov, continent-level

(c) Flu, country-level

(d) Cov, country-level

Figure 16: Negative log-likelihood (NLL) and reverse negative log-likelihood for FLU and COV on the *worst* sub-population. Lower is better. Error bands represent the 95% confidence interval across different oracle models. The worst sub-population for each model corresponds to the one with the highest average NLL at a temperature of 1.

| Model | Temperature | NLL | reverse NLL (95% CI) |
|---|---|---|---|
| Global | 0.2 | 0.1921 | 0.0099 (0.0096, 0.0102) |
| Prepend | 0.2 | 0.2003 | 0.0097 (0.0095, 0.0099) |
| Add_Embed | 0.2 | 0.1951 | 0.0090 (0.0089, 0.0091) |
| Param_share | 0.2 | 0.2210 | 0.0239 (0.0234, 0.0244) |
| LoRA | 0.2 | 0.1914 | 0.0099 (0.0097, 0.0101) |
| Finetune | 0.2 | 0.1964 | 0.0089 (0.0088, 0.0091) |
| **Transmission** | 0.2 | 0.1951 | 0.0095 (0.0091, 0.0099) |
| Global | 0.4 | 0.0967 | 0.0105 (0.0103, 0.0107) |
| Prepend | 0.4 | 0.1010 | 0.0100 (0.0097, 0.0102) |
| Add_Embed | 0.4 | 0.0984 | 0.0098 (0.0096, 0.0099) |
| Param_share | 0.4 | 0.1113 | 0.0250 (0.0247, 0.0253) |
| LoRA | 0.4 | 0.0964 | 0.0105 (0.0104, 0.0107) |
| Finetune | 0.4 | 0.0990 | 0.0097 (0.0095, 0.0098) |
| **Transmission** | 0.4 | 0.0981 | 0.0097 (0.0094, 0.0100) |
| Global | 0.6 | 0.0653 | 0.0124 (0.0122, 0.0126) |
| Prepend | 0.6 | 0.0684 | 0.0113 (0.0110, 0.0115) |
| Add_Embed | 0.6 | 0.0666 | 0.0114 (0.0112, 0.0115) |
| Param_share | 0.6 | 0.0752 | 0.0257 (0.0254, 0.0259) |
| LoRA | 0.6 | 0.0651 | 0.0124 (0.0122, 0.0125) |
| Finetune | 0.6 | 0.0670 | 0.0113 (0.0110, 0.0115) |
| **Transmission** | 0.6 | 0.0661 | 0.0105 (0.0102, 0.0108) |
| Global | 0.8 | 0.0503 | 0.0186 (0.0183, 0.0189) |
| Prepend | 0.8 | 0.0527 | 0.0152 (0.0150, 0.0154) |
| Add_Embed | 0.8 | 0.0515 | 0.0158 (0.0156, 0.0160) |
| Param_share | 0.8 | 0.0576 | 0.0274 (0.0271, 0.0277) |
| LoRA | 0.8 | 0.0502 | 0.0186 (0.0183, 0.0189) |
| Finetune | 0.8 | 0.0517 | 0.0158 (0.0155, 0.0161) |
| **Transmission** | 0.8 | 0.0505 | 0.0132 (0.0130, 0.0134) |
| Global | 1.0 | 0.0424 | 0.0382 (0.0373, 0.0391) |
| Prepend | 1.0 | 0.0445 | 0.0264 (0.0260, 0.0268) |
| Add_Embed | 1.0 | 0.0437 | 0.0282 (0.0275, 0.0289) |
| Param_share | 1.0 | 0.0479 | 0.0324 (0.0321, 0.0328) |
| LoRA | 1.0 | 0.0424 | 0.0385 (0.0376, 0.0395) |
| Finetune | 1.0 | 0.0436 | 0.0300 (0.0291, 0.0308) |
| **Transmission** | 1.0 | 0.0421 | 0.0244 (0.0236, 0.0252) |

Table 8: Average negative log-likelihood (NLL) and reverse negative log-likelihood for FLU at the continent level. Lower is better. The 95% confidence intervals (CI) across different oracle models are illustrated.

| Model | Temperature | NLL | reverse NLL (95% CI) |
|---|---|---|---|
| Global | 0.2 | 0.1880 | 0.0160 (0.0154, 0.0165) |
| Prepend | 0.2 | 0.2128 | 0.0133 (0.0126, 0.0140) |
| Add_Embed | 0.2 | 0.1895 | 0.0138 (0.0131, 0.0145) |
| Param_share | 0.2 | 0.1964 | 0.0156 (0.0131, 0.0182) |
| LoRA | 0.2 | 0.1880 | 0.0160 (0.0154, 0.0166) |
| Finetune | 0.2 | 0.1911 | 0.0142 (0.0136, 0.0148) |
| **Transmission** | 0.2 | 0.1663 | 0.0140 (0.0129, 0.0151) |
| Global | 0.4 | 0.0968 | 0.0167 (0.0161, 0.0172) |
| Prepend | 0.4 | 0.1093 | 0.0140 (0.0134, 0.0147) |
| Add_Embed | 0.4 | 0.0976 | 0.0147 (0.0140, 0.0154) |
| Param_share | 0.4 | 0.1002 | 0.0158 (0.0138, 0.0178) |
| LoRA | 0.4 | 0.0967 | 0.0167 (0.0161, 0.0172) |
| Finetune | 0.4 | 0.0984 | 0.0152 (0.0147, 0.0158) |
| **Transmission** | 0.4 | 0.0857 | 0.0154 (0.0144, 0.0165) |
| Global | 0.6 | 0.0670 | 0.0191 (0.0185, 0.0196) |
| Prepend | 0.6 | 0.0755 | 0.0156 (0.0149, 0.0162) |
| Add_Embed | 0.6 | 0.0679 | 0.0167 (0.0160, 0.0175) |
| Param_share | 0.6 | 0.0691 | 0.0167 (0.0151, 0.0183) |
| LoRA | 0.6 | 0.0670 | 0.0189 (0.0184, 0.0195) |
| Finetune | 0.6 | 0.0682 | 0.0172 (0.0167, 0.0178) |
| **Transmission** | 0.6 | 0.0599 | 0.0177 (0.0167, 0.0188) |
| Global | 0.8 | 0.0531 | 0.0245 (0.0239, 0.0251) |
| Prepend | 0.8 | 0.0595 | 0.0188 (0.0181, 0.0195) |
| Add_Embed | 0.8 | 0.0540 | 0.0207 (0.0199, 0.0215) |
| Param_share | 0.8 | 0.0545 | 0.0186 (0.0173, 0.0199) |
| LoRA | 0.8 | 0.0531 | 0.0243 (0.0237, 0.0249) |
| Finetune | 0.8 | 0.0540 | 0.0214 (0.0209, 0.0220) |
| **Transmission** | 0.8 | 0.0480 | 0.0217 (0.0207, 0.0227) |
| Global | 1.0 | 0.0459 | 0.0347 (0.0339, 0.0354) |
| Prepend | 1.0 | 0.0509 | 0.0258 (0.0249, 0.0267) |
| Add_Embed | 1.0 | 0.0469 | 0.0288 (0.0278, 0.0299) |
| Param_share | 1.0 | 0.0467 | 0.0227 (0.0218, 0.0236) |
| LoRA | 1.0 | 0.0459 | 0.0344 (0.0336, 0.0351) |
| Finetune | 1.0 | 0.0466 | 0.0302 (0.0294, 0.0310) |
| **Transmission** | 1.0 | 0.0420 | 0.0297 (0.0285, 0.0308) |

Table 9: Average negative log-likelihood (NLL) and reverse negative log-likelihood for Cov at the continent level. Lower is better. The 95% confidence intervals (CI) across different oracle models are illustrated.

| Model | Temperature | NLL | reverse NLL (95% CI) |
|---|---|---|---|
| Global | 0.2 | 0.2061 | 0.0146 (0.0142, 0.0150) |
| Prepend | 0.2 | 0.2294 | 0.0103 (0.0094, 0.0111) |
| Add_Embed | 0.2 | 0.2243 | 0.0097 (0.0092, 0.0102) |
| Param_share | 0.2 | 0.2494 | 0.0179 (0.0172, 0.0185) |
| LoRA | 0.2 | 0.2058 | 0.0142 (0.0138, 0.0146) |
| Finetune | 0.2 | 0.2162 | 0.0111 (0.0107, 0.0115) |
| **Transmission(G2G)** | 0.2 | 0.2113 | 0.0098 (0.0089, 0.0107) |
| **Transmission(G2L)** | 0.2 | 0.2213 | 0.0101 (0.0094, 0.0108) |
| Global | 0.4 | 0.1038 | 0.0151 (0.0148, 0.0155) |
| Prepend | 0.4 | 0.1155 | 0.0106 (0.0099, 0.0114) |
| Add_Embed | 0.4 | 0.1131 | 0.0106 (0.0102, 0.0110) |
| Param_share | 0.4 | 0.1256 | 0.0185 (0.0180, 0.0191) |
| LoRA | 0.4 | 0.1036 | 0.0146 (0.0142, 0.0149) |
| Finetune | 0.4 | 0.1089 | 0.0117 (0.0113, 0.0122) |
| **Transmission(G2G)** | 0.4 | 0.1063 | 0.0100 (0.0092, 0.0108) |
| **Transmission(G2L)** | 0.4 | 0.1112 | 0.0104 (0.0097, 0.0110) |
| Global | 0.6 | 0.0702 | 0.0177 (0.0174, 0.0181) |
| Prepend | 0.6 | 0.0780 | 0.0123 (0.0116, 0.0131) |
| Add_Embed | 0.6 | 0.0765 | 0.0124 (0.0121, 0.0127) |
| Param_share | 0.6 | 0.0847 | 0.0199 (0.0195, 0.0204) |
| LoRA | 0.6 | 0.0700 | 0.0171 (0.0168, 0.0174) |
| Finetune | 0.6 | 0.0736 | 0.0137 (0.0133, 0.0141) |
| **Transmission(G2G)** | 0.6 | 0.0716 | 0.0111 (0.0105, 0.0117) |
| **Transmission(G2L)** | 0.6 | 0.0749 | 0.0113 (0.0108, 0.0118) |
| Global | 0.8 | 0.0540 | 0.0275 (0.0272, 0.0278) |
| Prepend | 0.8 | 0.0598 | 0.0170 (0.0162, 0.0177) |
| Add_Embed | 0.8 | 0.0588 | 0.0164 (0.0163, 0.0165) |
| Param_share | 0.8 | 0.0648 | 0.0230 (0.0227, 0.0233) |
| LoRA | 0.8 | 0.0539 | 0.0272 (0.0269, 0.0275) |
| Finetune | 0.8 | 0.0566 | 0.0207 (0.0204, 0.0210) |
| **Transmission(G2G)** | 0.8 | 0.0549 | 0.0148 (0.0143, 0.0152) |
| **Transmission(G2L)** | 0.8 | 0.0573 | 0.0148 (0.0144, 0.0152) |
| Global | 1.0 | 0.0455 | 0.0626 (0.0620, 0.0632) |
| Prepend | 1.0 | 0.0499 | 0.0294 (0.0282, 0.0306) |
| Add_Embed | 1.0 | 0.0491 | 0.0291 (0.0290, 0.0293) |
| Param_share | 1.0 | 0.0537 | 0.0302 (0.0299, 0.0305) |
| LoRA | 1.0 | 0.0453 | 0.0594 (0.0589, 0.0600) |
| Finetune | 1.0 | 0.0474 | 0.0465 (0.0463, 0.0467) |
| **Transmission(G2G)** | 1.0 | 0.0458 | 0.0337 (0.0335, 0.0339) |
| **Transmission(G2L)** | 1.0 | 0.0476 | 0.0314 (0.0312, 0.0316) |

Table 10: Average negative log-likelihood (NLL) and reverse negative log-likelihood for FLU at the country level. Lower is better. The 95% confidence intervals (CI) across different oracle models are illustrated.

| Model | Temperature | NLL | reverse NLL (95% CI) |
|---|---|---|---|
| Global | 0.2 | 0.2103 | 0.0150 (0.0148, 0.0152) |
| Prepend | 0.2 | 0.2293 | 0.0165 (0.0164, 0.0166) |
| Add_Embed | 0.2 | 0.3318 | 0.0157 (0.0155, 0.0158) |
| Param_share | 0.2 | 0.2478 | 0.0403 (0.0396, 0.0411) |
| LoRA | 0.2 | 0.2104 | 0.0151 (0.0149, 0.0153) |
| Finetune | 0.2 | 0.2129 | 0.0150 (0.0148, 0.0152) |
| **Transmission(G2G)** | 0.2 | 0.1984 | 0.0144 (0.0142, 0.0146) |
| **Transmission(G2L)** | 0.2 | 0.1923 | 0.0146 (0.0145, 0.0147) |
| Global | 0.4 | 0.1079 | 0.0167 (0.0165, 0.0169) |
| Prepend | 0.4 | 0.1173 | 0.0169 (0.0168, 0.0169) |
| Add_Embed | 0.4 | 0.1682 | 0.0160 (0.0159, 0.0161) |
| Param_share | 0.4 | 0.1261 | 0.0388 (0.0382, 0.0395) |
| LoRA | 0.4 | 0.1079 | 0.0167 (0.0166, 0.0169) |
| Finetune | 0.4 | 0.1092 | 0.0165 (0.0163, 0.0167) |
| **Transmission(G2G)** | 0.4 | 0.1013 | 0.0152 (0.0150, 0.0153) |
| **Transmission(G2L)** | 0.4 | 0.0983 | 0.0151 (0.0151, 0.0152) |
| Global | 0.6 | 0.0749 | 0.0192 (0.0190, 0.0194) |
| Prepend | 0.6 | 0.0809 | 0.0179 (0.0178, 0.0179) |
| Add_Embed | 0.6 | 0.1146 | 0.0169 (0.0168, 0.0170) |
| Param_share | 0.6 | 0.0865 | 0.0379 (0.0373, 0.0384) |
| LoRA | 0.6 | 0.0750 | 0.0191 (0.0190, 0.0193) |
| Finetune | 0.6 | 0.0759 | 0.0187 (0.0185, 0.0188) |
| **Transmission(G2G)** | 0.6 | 0.0698 | 0.0166 (0.0165, 0.0167) |
| **Transmission(G2L)** | 0.6 | 0.0678 | 0.0164 (0.0163, 0.0165) |
| Global | 0.8 | 0.0596 | 0.0233 (0.0232, 0.0235) |
| Prepend | 0.8 | 0.0637 | 0.0200 (0.0199, 0.0201) |
| Add_Embed | 0.8 | 0.0886 | 0.0188 (0.0188, 0.0189) |
| Param_share | 0.8 | 0.0675 | 0.0380 (0.0375, 0.0385) |
| LoRA | 0.8 | 0.0596 | 0.0232 (0.0231, 0.0234) |
| Finetune | 0.8 | 0.0603 | 0.0225 (0.0224, 0.0227) |
| **Transmission(G2G)** | 0.8 | 0.0549 | 0.0194 (0.0193, 0.0194) |
| **Transmission(G2L)** | 0.8 | 0.0534 | 0.0189 (0.0188, 0.0189) |
| Global | 1.0 | 0.0517 | 0.0315 (0.0314, 0.0315) |
| Prepend | 1.0 | 0.0545 | 0.0252 (0.0249, 0.0254) |
| Add_Embed | 1.0 | 0.0739 | 0.0234 (0.0233, 0.0235) |
| Param_share | 1.0 | 0.0569 | 0.0402 (0.0397, 0.0406) |
| LoRA | 1.0 | 0.0517 | 0.0310 (0.0310, 0.0310) |
| Finetune | 1.0 | 0.0522 | 0.0298 (0.0297, 0.0299) |
| **Transmission(G2G)** | 1.0 | 0.0468 | 0.0273 (0.0272, 0.0274) |
| **Transmission(G2L)** | 1.0 | 0.0458 | 0.0279 (0.0278, 0.0281) |

Table 11: Average negative log-likelihood (NLL) and reverse negative log-likelihood for Cov at the country level. Lower is better. The 95% confidence intervals (CI) across different oracle models are illustrated.

