# OpenReview forum: "Predicting sub-population specific viral evolution"
_TMLR — Accepted by TMLR_

### Review · Reviewer_QJzL · 2024-11-23

**Summary Of Contributions:**

This manuscript introduces a deep time-series graph neural ordinary differential equation model for predicting the distribution of viral proteins in different sub-populations, aiming to improve upon traditional finite element method simulations in multi-physics field predictions during the curing process of laminated composites.

**Audience:**

Yes

**Claims And Evidence:**

Yes

**Requested Changes:**

See Weaknesses.

**Strengths And Weaknesses:**

Strengths:
- The integration of deep learning with differential equations to model viral evolution presents a novel approach.
- The model's ability to capture the dynamic interactions between sub-populations is important especially for diseases with complex transmission patterns.

Weaknesses:
- To what extent does the model account for the potential non-linear relationships between transmission rates and viral protein distributions?
- Could the model's reliance on hierarchical structures lead to overfitting, particularly when the inherent structures relating sub-populations are not accurately represented?
- What steps are taken to ensure that the hierarchical variation of the model, which re-parameterizes the transmission rate matrix, does not lose critical information when reducing the complexity of the transmission dynamics?
- How does the model account for the potential variability in viral mutation rates, which could significantly impact the accuracy of protein distribution predictions?
- Can the authors elaborate on the model's sensitivity to changes in the underlying assumptions about viral transmission pathways, and how this might impact the model's predictive accuracy over time?
- What steps are taken to ensure that the hierarchical variation of the model, which re-parameterizes the transmission rate matrix, does not lose critical information when reducing the complexity of the transmission dynamics?

---

> ### Author Response · Authors · 2024-12-11
>
> We appreciate your feedback. Below, we address the specific weaknesses and requested clarifications to improve the manuscript.
>
> > To what extent does the model account for the potential non-linear relationships between transmission rates and viral protein distributions?
>
> In the epidemiology literature, the linear relationship between transmission rate and distribution is considered state-of-the-art [1-3]. Following this practice, our model also assumes a linear relationship (Equation 1). From a computational perspective, this assumption facilitates analytical tractability by enabling a closed-form solution to the underlying ODE while maintaining strong empirical performance in capturing distributional changes. The linear formulation is particularly advantageous for scalability, allowing the model to handle large datasets and support the likelihood calculation at any given time.
>
> While incorporating non-linear dynamics could potentially enhance performance, extending the model to include such relationships would require the use of numerical ODE solvers, which are computationally intensive—particularly since we need to solve the ODE for each training sample. We view this as a promising direction for future work and will highlight it in the revised manuscript.
>
> [1] Kermack, W. O., & McKendrick, A. G. (1927). A contribution to the mathematical theory of epidemics. Proceedings of the royal society of london. Series A, Containing papers of a mathematical and physical character, 115(772), 700-721.
>
> [2] Łuksza, M., & Lässig, M. (2014). A predictive fitness model for influenza. Nature, 507(7490), 57-61.
>
> [3] Obermeyer, F., Jankowiak, M., Barkas, N., Schaffner, S. F., Pyle, J. D., Yurkovetskiy, L., ... & Lemieux, J. E. (2022). Analysis of 6.4 million SARS-CoV-2 genomes identifies mutations associated with fitness. Science, 376(6599), 1327-1332.
>
> > Could the model's reliance on hierarchical structures lead to overfitting, particularly when the inherent structures relating sub-populations are not accurately represented?
>
> Our empirical results show no evidence of overfitting due to the reliance on hierarchical structures. As demonstrated in our ablation study (Figure 6), even when random grouping is applied, the model continues to outperform the best baseline model.
>
> > What steps are taken to ensure that the hierarchical variation of the model, which re-parameterizes the transmission rate matrix, does not lose critical information when reducing the complexity of the transmission dynamics?
>
> The re-parameterization of the transmission rate matrix preserves the same representation capacity as the original matrix $A$ if $A$ is diagonalizable. If $A$ is diagonalizable, it is block-diagonalizable. That means for any diagonalizable transmission matrix, we can find a matrix $W$ and a block-diagonal matrix $\overline{A}$ such that $A=W\overline{A}W^{-1}$. We have included a formal proof of this in Appendix A.2 of the revised manuscript.
>
> While the G2G and G2L approximations simplify the transmission model, this reduction in complexity does not compromise performance. Our ablation study (Figure 7) demonstrates that these approximations yield the best results, validating the effectiveness of this assumption.
>
>
> > How does the model account for the potential variability in viral mutation rates, which could significantly impact the accuracy of protein distribution predictions?
>
> The model implicitly accounts for variability in viral mutation rates by learning from historical epidemiological data. Specifically, the neural network learns to capture the diverse growth patterns of different variants across specific locations in the transmission rate matrix.
>
> > Can the authors elaborate on the model's sensitivity to changes in the underlying assumptions about viral transmission pathways, and how this might impact the model's predictive accuracy over time?
>
> Currently, the model implicitly learns viral transmission pathways from the data. However, if reliable information about the transmission patterns (e.g., transportation rates between locations) is available, it can be incorporated as a prior to enhance the model's predictive accuracy. For instance, in Equation 1, such information could be integrated directly into the transmission rate matrix as an additional constraint or regularization term.

---

### Review · Reviewer_W9g8 · 2024-11-24

**Summary Of Contributions:**

The authors create a biologically inspired machine learning model to predict how viral proteins will evolve over space and time.

**Audience:**

Yes

**Broader Impact Concerns:**

On a darker note, model could also be used to generate sequences that are highly novel in a subpopulation that are likely to be transmissible, and therefore more deadly.

**Claims And Evidence:**

Yes

**Requested Changes:**

“Throughout this paper, we use location interchangeably with sub-population.” I feel like this is a bit confusing. I had originally thought that ‘location’ meant ‘site’, as in the site of mutation within the protein. Sub-population is seemingly the cleanest and most technically correct.

“ The model outputs Nt(x), the un-normalized probability (occurrence) of x at time t in all locations: Nt(x) = [nt(x; 1), ..., nt(x; m)].” What is m?

“ Intuitively, [Aθ(x)]ij measures the number of people in location i infected by one person from location j during dt.” What is j?

“...outputs |V | transmission rate matrices…” Why the ‘|’ character around the V?

Because this model is autoregressive, it will generate L different transmission matrices per protein? This isn’t explicitly stated between equations 4 and 5.

“We obtain pt(xs|x<s) by normalizing nt(xs, x<s) over all possible values of xs.” Do you actually normalize over all possible values, or do you just use the autoregressive likelihood.

**Strengths And Weaknesses:**

Strengths

I think the formulation of the problem is good. It is clear the authors are familiar with the biology  of the domain, as well as the approaches used in the field.

Taking advantage of the geographic structure is very interesting and reasonable for some aspects of this work. With the amount of intercontinental travel, do variants stay geographically isolated for long periods of time?

“Since egg-based vaccines require lead times of up to 6 months, we train our models on data collected before February of each year, and evaluate models on the sequences collected from October to March of the next year (winter season),” - This is a good evaluation!

“We generate 50 and 500 sequences with the highest probabilities using beam search (Sutskever et al., 2014) and calculate their coverage, defined as the ground-truth frequencies of these sequences occurring in the testing time and locations.” This is also a very interesting way of evaluating performance.
.
The different model ablations and modifications (Global, finetune, prepend, add embed, and parameter sharing) are also very interesting.

“To ensure a fair comparison, we re-trained their protein language models on our training set.” This is great too, and probably a lot of work.

Weaknesses

The notation is messy and all terms should be defined more clearly in the initial description of terms.

“Since eigendecomposition scales as O(m3 ) with the number of locations, applying our model directly to fine-grained locations such as countries becomes highly time-consuming…” Are you actually doing any eigendecomposition with this model?I thought it was parameterized by a neural network? If so, why even mention this? Or does the model output some data, and then you do eigendecomposition?

---

> ### Author Response · Authors · 2024-12-11
>
> Thank you for your valuable and detailed feedback. The manuscript has been updated to streamline the notation as you requested. Below, we address the specific points you raised:
>
> > Do variants stay geographically isolated for long periods of time?
>
> Due to global travel, viral variants typically spread to other geographical locations within a few months. However, the distribution of variants often remains distinct across regions. This distinction arises from the local-specific new variants and the asynchronous transmission of existing ones. This phenomenon is nicely visualized by this publicly available tool https://nextstrain.org/seasonal-flu/h3n2/ha/2y (map in the visualization).
>
>
> >Are you actually doing any eigendecomposition with this model?I thought it was parameterized by a neural network? If so, why even mention this? Or does the model output some data, and then you do eigendecomposition?
>
> We applied the eigendecomposition to the outputs of neural networks. Eigendecomposition is required to obtain the closed-form solution of the ODE that models the evolution of probabilities, as illustrated in Equation 2 and Appendix A.1.
> Specifically, the neural network (a GTP-2 variant) outputs a transmission matrix $A(x_s,x_{<s})$ for each residue in input sequence $x$. We then perform eigendecomposition on $A$ using `torch.eigh` to obtain its eigenvalues and eigenvectors. After that, we calculate the $p_t(x_s|x_{<s})$, based on the eigenvalues and eigenvectors.
>
> > “Throughout this paper, we use location interchangeably with sub-population.” I feel like this is a bit confusing. I had originally thought that ‘location’ meant ‘site’, as in the site of mutation within the protein. Sub-population is seemingly the cleanest and most technically correct.
>
> Thank you for pointing this out. To avoid confusion, we have clarified the definition of “location”, explicitly specifying that it refers to geographical locations throughout the paper.
>
> > “ The model outputs Nt(x), the un-normalized probability (occurrence) of x at time t in all locations: Nt(x) = [nt(x; 1), ..., nt(x; m)].” What is m?
>
> $m$ is the total number of sub-populations, clarified on Page 2.
>
> > “ Intuitively, [Aθ(x)]ij measures the number of people in location i infected by one person from location j during dt.” What is j?
>
> $j$ refers to the column index of the entry $(i, j)$ in the matrix $A_\theta(x)$. Intuitively, $A_{ij}$ captures the transmission rate from location $j$ to location $i$.
>
> > “...outputs |V | transmission rate matrices…” Why the ‘|’ character around the V?
>
> $|V|$ indicates the size/cardinality of set $V$, which represents the vocabulary of all possible amino acids.
>
> > Because this model is autoregressive, it will generate L different transmission matrices per protein? This isn’t explicitly stated between equations 4 and 5.
>
> Thank you for your suggestion. We have added an explicit explanation between equations 4 and 5 to clarify this.
>
> > “We obtain pt(xs|x<s) by normalizing nt(xs, x<s) over all possible values of xs.” Do you actually normalize over all possible values, or do you just use the autoregressive likelihood.
>
> We used the auto-regressive likelihood. Here, $x_s$ is the $s$-th residue in the protein sequences, which can take on ~20 possible amino acid categories. This allows us to normalize the $p_t(x_s|x_{<s})$ directly.
>
> **Broader Impact Concerns**
>
> We appreciate the broader impact concerns you have raised. In principle, while it is possible for models to generate highly novel variants, these variants are typically out-of-distribution relative to existing data, making their generation by our model unlikely. Furthermore, the synthesis and testing of viruses are strictly regulated and controlled by relevant authorities, thereby minimizing potential misuse.

---

### Review · Reviewer_2RdL · 2025-01-07

**Summary Of Contributions:**

The manuscript presents an approach to model the propagation of virus (amino-acid sequence) strains across both time and sub-populations (continents, countries). While common approaches model the evolution of such viruses either as a whole or independently for each subpopulation, they present methodology that allows to model the propagation in aggregate across subpopulations while maintaining the ability to predict fine-grained prediction for each country. They achieve this through hierarchical modeling of earth because of the intractability of modeling interactions across each pair of sub-populations. They show state-of-the-art results on the Pareto-frontier in the negative log-likelihood and reverse negative log-likelihood space compared to evaluated baselines.

**Audience:**

Yes

**Broader Impact Concerns:**

I see no broader impact concerns.

**Claims And Evidence:**

Yes

**Requested Changes:**

* Could you add a description of what is a protein sequence residue in section 2.1?
* In section 2.2, I think a comment about the specific limitations of group-to-location and group-to-group setting would be useful here to understand the compute/expressivity tradeoff. e.g., is the a scenario where these simplification would be incapable of modeling a location-to-location interaction and to which degree?
* In the ablation study (section 4.4), could you comment on why is the no hierarchy model worse than the G2G clustering model? Shouldn’t it be at least as expressive?
* A common measure for sub-population distribution shift problem is performance on worst subpopulation. Are there settings here in the results where agglomerated approaches (global, or param sharing) that outperform the transmission approach on their worst subpopulation.
* Could you elaborate a bit further on how were groups chosen in table 6 and 7? Specifically why not make more than 2 groups for the Flu model and have something closer to the Cov groups?
* I see the code has been provided in the supplementary material. Would be good to have a sentence in the main text pointing to a repository (Placeholder for now)

**nit**
* Some of the plots are readable, but a bit hard to read (some points are obscured by others). Perhaps something could be do for that, but not a hard requirement

**Strengths And Weaknesses:**

**Strengths**
* The paper is well polished. It reads clearly and figures are a great support for detailing the moving parts.
* The mathematical formulation of the methodology is clear and rigorous and proofs are provided for the necessary parts (thanks to recent revisions).
* Results are strong, clearly outperforms baselines.
* Ablation results are well constructed and creates trust in the methodology.
* Have strong notion of their own limitation (as described in the conclusion).

**Weaknesses**
* Related works section seems a bit narrow.
* Few of the baselines are published, notably for baselines that incorporate location information.
* I see a bit of a parallel/opportunity here with the Neural Ordinary Differential Equations field of research which might provide useful modeling tool for this approach in terms of precision which has not been mentioned here.

[1] Chen, Ricky TQ, et al. "Neural ordinary differential equations." Advances in neural information processing systems 31 (2018).

---

> ### Author Response · Authors · 2025-01-16
>
> Thank you for your valuable feedback. The manuscript has been updated according to your suggestions. Below, we address the specific points you raised:
>
> > Related works section seems a bit narrow.
>
> We have expanded the related works section to include a more thorough discussion about Neural ODEs and SDEs [1-4].
>
> [1] Chen, R. T., Rubanova, Y., Bettencourt, J., & Duvenaud, D. K. (2018). Neural ordinary differential equations. Advances in neural information processing systems, 31.
>
> [2] Campbell, A., Benton, J., De Bortoli, V., Rainforth, T., Deligiannidis, G., & Doucet, A. (2022). A continuous time framework for discrete denoising models. Advances in Neural Information Processing Systems, 35, 28266-28279.
>
> [3] Lipman, Y., Chen, R. T., Ben-Hamu, H., Nickel, M., & Le, M. Flow Matching for Generative Modeling. In The Eleventh International Conference on Learning Representations.
>
> [4] Song, Y., Sohl-Dickstein, J., Kingma, D. P., Kumar, A., Ermon, S., & Poole, B. Score-Based Generative Modeling through Stochastic Differential Equations. In International Conference on Learning Representations.
>
> > Few of the baselines are published, notably for baselines that incorporate location information.
>
> While there exist baselines for general protein evolution (EVEscape, VaxSeer), this is the first machine learning work that addresses the task of sub-population protein evolution. Thus, the most relevant published baselines do not incorporate location, and the baselines that consider location are our own ablation studies.
>
> > Applying Neural ODE
>
> The Neural ODE is not directly applicable to our task since the variable evolving in our model is the distribution of discrete sequences, rather than continuous representations. Defining a more general ODE, like Neural ODE, can potentially enhance the capacity of our approach. However, the need for an ODE solver would introduce inefficiencies in training and hinder scalability to larger datasets. We have included this discussion in both the Related Work and Discussion sections, as you suggested.
>
> > Could you add a description of what is a protein sequence residue in section 2.1?
>
> In the context of protein sequences, a residue is a single "token" or "word", that refers to a single “amino acid” building block of a protein. We have updated the text to clarify this point in section 2.1.
>
> > In section 2.2, I think a comment about the specific limitations of group-to-location and group-to-group setting would be useful here to understand the compute/expressivity tradeoff. e.g., is the a scenario where these simplification would be incapable of modeling a location-to-location interaction and to which degree?
>
> The extent to which the G2L and G2G models have/lack the expressivity to model location-to-location transmissions depends on the complexity of viral transmissions across the globe. We believe these approximations are reasonable in practice, as transmission between distant countries is often dominated by some major routines. For example, when traveling from an Asian country to the United States, the journey may involve a stop at a major port in Asia before reaching the destination (group-to-location), or it may involve travel between two major ports across continents before reaching the final destination (group-to-group).
>
> On the other hand, in an extremely connected world where some sub-populations exhibit significantly distinct transmission patterns, these simplified models may fall short of representing such complexities. This issue can be mitigated by refining the group configurations to separate such sub-populations better. Our empirical studies show that these approximations perform robustly well overall. We have added this discussion, along with the intuitive explanation, to Section 2.2.
>
> > In the ablation study (section 4.4), could you comment on why is the no hierarchy model worse than the G2G clustering model? Shouldn’t it be at least as expressive?
>
> We hypothesize that the better performance of the G2G/clustering model is due to the additional information introduced by clustering locations based on their geographical coordinates. In contrast, the G2G model with random groups performs similarly to the no hierarchy model, highlighting the benefit of incorporating geographical information.
>
> > A common measure for sub-population distribution shift problem is performance on worst subpopulation. Are there settings here in the results where agglomerated approaches (global, or param sharing) that outperform the transmission approach on their worst subpopulation.
>
> Our method consistently outperforms all other methods on the worst-case subgroups. We illustrated the NLL and reverse NLLs of all models on their worst subpopulations in Appendix Figure 16. Notably, our model still attains the best frontier.

---

> > ### Author Response · Authors · 2025-01-16
> >
> > > Could you elaborate a bit further on how were groups chosen in table 6 and 7? Specifically why not make more than 2 groups for the Flu model and have something closer to the Cov groups?
> >
> > Due to computational resource constraints, we limited the size of each group to ensure that training could be completed within a few days (typically fewer than 16 countries per group). For Flu, since only 16 countries had sufficient data in the training set (Appendix Table 2), we chose 2 groups. For Cov, we used 3 groups because there are 32 countries with sufficient data. In the ablation study (Figure 7), we experimented with various group configurations to show that the performance is robust to the choice of the number of groups.
> >
> > > Repository placeholder
> >
> > Thank you for your suggestion. We have added a repository placeholder in the Abstract, and the codebase will be available in the later non-anonymous version.
> >
> >
> > > Some of the plots are readable, but a bit hard to read (some points are obscured by others). Perhaps something could be do for that, but not a hard requirement
> >
> > To improve readability, the exact values of NLL and reverse NLL for Fig. 4 are provided in Appendix Tab.8-11 in the revised version.

---

> > > ### Comment · Reviewer_2RdL · 2025-02-03
> > > **Addressing rebuttal**
> > >
> > > Thanks to the author of the manuscript for answering my concerns.
> > >
> > > Everything has been addressed adequately.

---

### Author Response · Authors · 2025-01-16

We appreciate all the reviewers’ suggestions, and have made the following updates to our paper:

- We included Figure 16 in the Appendix to show that our method performs better in the worst subpopulations.
- We added the Appendix Table. 8-11 providing the exact number in Figure 4 for better readability.
- We added an intuitive explanation of G2G/G2L approximation with a discussion about its limitations in Section 2.2.
- We extended the related work section to include a discussion on Neural ODE/SDEs.
- We added a repository placeholder in the Abstract, and the codebase will be publicly available in a future non-anonymous version.

The new revision is highlighted in orange.

---

### Decision · Action_Editor_VdQ5 · 2025-02-04

**Recommendation:** Accept as is

**Comment:**

This is a well-presented and throughout solid paper that shows an interesting model with thorough empirical evaluation. All reviews are positive. The paper presentation is sufficient for accepting as is. This paper fits well with TMLR.

**Audience:**

The topic of viral evolution is applied, but is a problem of spatiotemporal sequence distributions, which have general applicability and interest to ML audience. The paper also parallels the work of neural ODEs, which is likely to induce further ML research into the topic. All reviewers agree.

**Claims And Evidence:**

The paper models spatially distributed viral sequence distributions through a neural network parameterised linear ODE. The sub-population level modelling is novel in this context, and model's performance is supported by empirical evidence. All reviewers agree.